# ProtCID: a data resource for structural information on protein interactions

Qifang Xu [1] & Roland L. Dunbrack Jr. [1]*

Structural information on the interactions of proteins with other molecules is plentiful, and for some proteins and protein families, there may be 100s of available structures. It can be very difficult for a scientist who is not trained in structural bioinformatics to access this information comprehensively. Previously, we developed the Protein Common Interface Database (ProtCID), which provided clusters of the interfaces of full-length protein chains as a means of identifying biological assemblies. Because proteins consist of domains that act as modular functional units, we have extended the analysis in ProtCID to the individual domain level. This has greatly increased the number of large protein-protein clusters in ProtCID, enabling the generation of hypotheses on the structures of biological assemblies of many systems. The analysis of domain families allows us to extend ProtCID to the interactions of domains with peptides, nucleic acids, and ligands. ProtCID provides complete annotations and coordinate sets for every cluster.

[1] Fox Chase Cancer Center, 333 Cottman Avenue, Philadelphia, PA 19111, USA. *email: Roland.Dunbrack@fccc.edu

All proteins function via interactions with other molecules, including nucleic acids, small molecular ligands, ions, and other proteins—in the form of both homo-oligomers and hetero-oligomers. How such interactions occur and defining their role in protein function are the central goals of structural biology. For any protein system of interest, it is valuable to understand the structure in all known functional forms, including different conformations and oligomeric states and interactions with nucleic acids, ligands, and other proteins. To accomplish this, it is often necessary to examine 100s or even 1000s of available structures of a protein and its homologs in the Protein Data Bank (PDB)[1]. This process is very challenging and time consuming even for scientists trained in bioinformatics.

Of central importance to the utility of experimental structures is the accuracy of annotations. Authors of crystal structures are required to deposit a biological assembly into the PDB, which is what they believe to be the biologically relevant oligomeric form present in the crystal. This is in contrast to the asymmetric unit, which is the set of coordinates used to model the crystal lattice. The author-deposited biological assembly is different from the asymmetric unit for about 40% of crystal structures in the PDB[2]. Various authors have estimated the accuracy of the biological assemblies in the PDB in the range of 80–90%[3–6].

A common approach for identifying biological assemblies of molecules within protein crystals is to compare multiple crystal forms of the same or related proteins. Each crystal form will have different non-biological interfaces between proteins and with crystallization ligands, while in most cases the biological interactions will be shared between them. We have shown that if a homodimeric or heterodimeric interface is present in multiple crystal forms, especially when the proteins in the different crystals are homologous but not identical, then such interfaces are very likely to be part of biologically relevant assemblies[4].

To enable this form of analysis, we previously developed PDBfam[7], which assigns protein domain families (as defined by Pfam[8]) to every protein sequence in the PDB, and the Protein Common Interface Database (ProtCID), which compares and clusters the interfaces of pairs of full-length protein chains with defined Pfam domain architectures in different entries in the PDB[9]. ProtCID provides clusters of homodimeric and heterodimeric chain–chain interactions across PDB entries whether they are present within asymmetric units, between asymmetric units, or between unit cells. Although ProtCID was limited to protein–protein interfaces of full-length chains, it has been very useful in identifying biologically relevant interfaces and assemblies within crystals[10–13], including those that were not annotated in the PDB's biological assemblies. ProtCID provides coordinates and PyMol scripts for visualizing interfaces in each cluster.

Accessing the full amount of structural information in the PDB for a particular problem, a database needs to perform several tasks: (1) dividing proteins into conserved domain families, since many functions are performed by different types of domains and combined in different ways in different proteins; (2) creating coordinates of all protein–protein interactions for each PDB entry using crystal symmetry operators, since many biologically relevant interactions are not present in the asymmetric unit or in the PDB biological assemblies; (3) with these coordinates in hand, clustering of homo- and heterodimeric protein–protein and domain–domain interactions, and interactions of proteins domains with peptides, nucleic acids, and other ligands; (4) automated download access to the Cartesian coordinates so that visualization and analysis can be performed. Several servers analyze interfaces in either the asymmetric units and/or the biological assemblies of PDB entries[14–18], and some are intended to predict which interfaces may be biologically relevant from conservation scores and physical features and using machine learning predictors[5,19,20]. Interactions with peptides,

nucleic acids, and ligands have also been presented in several webservers and databases[21–27]. Very few of these resources provide clustering of structural information across the PDB, and very few provide coordinates for download.

In this paper, we extend the ProtCID approach from clustering full-length protein chains to clustering domain–domain interactions within protein crystals. The inclusion of interactions between individual domains greatly extends our ability to generate hypotheses about the functional interactions of proteins. We show examples of domain-level ProtCID clusters for some experimentally validated, biologically relevant protein–protein interactions that were in some way challenging to identify in the biological literature. This is especially true of weaker interactions within homooligomers, which are very difficult to distinguish from crystallization-induced interactions.

Analysis at the domain level has enabled new features in ProtCID: clustering of interactions of protein domains with peptides, nucleic acids, and small-molecule ligands. We have added access to ProtCID data at the level of protein superfamilies, which Pfam refers to as clans. It is often the case that there is no structure of a protein for a particular Pfam that contains biologically relevant information such as peptide, nucleic acid, or ligand binding. Thus, the existence in ProtCID of interactions for one protein family can be used to develop hypotheses for the structures of other protein families within the same superfamily. We show some examples of this approach. We have enabled a new search feature in ProtCID with which a user can identify possible domain–domain and domain/peptide interactions that are possible amongst a set of proteins (or between one hub protein and other proteins). This utility enables the identification of structural information on direct protein–protein interactions that might occur in large multi-subunit protein complexes.

ProtCID provides clustered structures of the interactions of protein domains with other protein domains, peptide, nucleic acids, and ligands. Each cluster is highly annotated, including PDB ids and chain IDs, protein names, UniProt ids, species, protein domain family identifiers (Pfams), crystal forms, surface areas, and biological assembly annotations. A crucial feature of ProtCID is the ability to download single archive files that contain coordinates for each interaction and scripts for visualizing the structures.

## Results

**ProtCID database and web site**. The ProtCID database contains information on four types of interactions: protein–protein interactions at the chain level, protein–protein interactions at the domain level, domain–peptide interactions, and the interactions of domains with nucleic acids and ligands. Domain–domain interactions can be between domains of the same Pfam or different Pfams, and can be interchain or intrachain. Chain–chain interactions are between chains with the same domain architecture (usually homodimers) or between those with different architectures.

Generating hypotheses for protein interactions by observing them in multiple crystals of homologous proteins requires grouping proteins in the PDB into homologous families at the domain and chain levels. To accomplish this, we utilize our database called PDBfam[7] containing 8636 Pfams observed within the PDB. Each PDB chain is annotated by a Pfam architecture as the ordered sequence of Pfams along the chain, e.g., (SH3)_(SH2)_(Pkinase). ProtCID contains 12,914 Pfam chain architectures from 42,336 proteins. Statistics on the number of the interactions, clusters, PDB entries, and Pfam domains are provided in Table 1 and in more detail in Supplementary Table 1. The table breaks protein–protein interactions down into four groups: Same chain-architectures

**Table 1 Summary of ProtCID interactions.**

|  |  | Chain-same[a] | Domain-same | Chain-diff[b] | Domain-diff | Pfam-peptide[c] | Pfam-DNA/RNA[d] | Pfam-ligand[e] |
|---|---|---|---|---|---|---|---|---|
| All | #Pfam-Archs | 5319 | 4461 | 3471 | 6571 | 1083 | 1260 | 6485 |
|  | #UniProts | 25,019 | 27,175 | 7738 | 15,335 | 2231 | 3952 | 26,950 |
| ≥2 UniProts | #Pfam-Archs | 2978 | 3029 | 2183 | 4304 | 319 | 632 | 3263 |
|  | #Clusters | 16,249 | 24,201 | 3359 | 9079 | 407 |  | 35,305 |
|  | #UniProts | 21,556 | 24,858 | 6850 | 14,361 | 1531 | 3600 | 22,931 |
| ≥5 UniProts | #Pfam-Archs | 847 | 1011 | 436 | 1298 | 78 | 254 | 1280 |
|  | #Clusters | 1775 | 2943 | 560 | 1893 | 88 |  | 5761 |
|  | #UniProts | 12,364 | 15,381 | 3511 | 9937 | 955 | 2983 | 16,835 |
| ≥10 UniProts | #Pfam-Archs | 339 | 458 | 166 | 523 | 35 | 156 | 557 |
|  | #Clusters | 536 | 813 | 196 | 666 | 37 |  | 1496 |
|  | #UniProts | 8196 | 10,802 | 2324 | 7143 | 706 | 2577 | 12,084 |
| ≥20 UniProts | #Pfam-Archs | 120 | 178 | 50 | 189 | 12 | 51 | 192 |
|  | #Clusters | 173 | 270 | 56 | 238 | 12 |  | 368 |
|  | #UniProts | 5010 | 6862 | 1425 | 4598 | 410 | 1485 | 7314 |

[a]"Same" refers to an interface between two chains or two domains with the same Pfam architecture. #Pfam-Archs is the number of chain-architecture pairs for chain-level (under "Chain") or Pfam-domain pairs for domain-level (under "Domain") in clusters that satisfy the rules in the first column (the minimum number of Uniprots listed and minimum sequence identity <90%). #Clusters is the number of clusters that satisfy the rules in the first column, and #UniProts is the number of unique UniProt codes in those clusters.
[b]"Diff" refers to an interface between two chains or two domains with different Pfam architectures.
[c]#Pfam-Archs is the number of Pfams which interact with peptides. A peptide is defined as a polypeptide chain with length less than 30 residues. #UniProts is the number of distinct protein-domain UniProts. Seqid is the minimum sequence identity of protein chains in a cluster.
[d]Pfam-DNA/RNA interactions are not clustered. #Pfam-Archs is the number of Pfams that follow the rules in the first column. No sequence identity cutoff is enforced.
[e]Pfam-ligand interactions do not use sequence identity cutoff. Any small molecule except water is considered a ligand. #Pfam-Archs is the number of Pfams which interact with any ligands given the number of Uniprots in the first column.

(column 3), same domain-architectures (column 4), different chain-architectures (column 5), different domain architectures (column 6).

The statistical advantage of analyzing protein–protein interactions at the domain level is evident in Table 1 by comparing column 3 with column 4 and by comparing column 5 with column 6. For example, for interactions between proteins with the same architecture, there are 1775 clusters with at least 5 Uniprot sequences at the chain level and 2943 clusters with at least 5 Uniprot sequences at the domain level. For clusters with different Pfam architectures, there are about three times as many clusters at the domain level than there are at the chain level. Currently, there are 1083 Pfams interacting with peptides, 1260 Pfams interacting with nucleic acids, and 6514 Pfams interacting with ligands.

On the ProtCID web site, there are four types of inputs: (i) a PDB ID; (ii) one or two Pfam IDs or accession codes; (iii) one or two protein sequences; (iv) one or more UniProt IDs. A user can browse Pfam IDs, Clan IDs, Pfam–Pfam pairs, peptide-interacting Pfams, ligands, and Pfam–Pfam networks (Fig. 1). Pfams are assigned to user-input sequences and UniProt IDs by HMMER3[28]. Except for PDB ID input, all inputs result in a list of PDB structures containing the Pfams, providing a comprehensive overview of homologous structures for any given query. From one structure, a user can check the clusters for chain–chain, domain–domain, and domain–peptide interactions. Coordinates and sequences are downloadable for each cluster including scripts for visualizing the interactions in PyMOL.

**Domain-domain interactions data in ProtCID.** Grouping proteins in the PDB into homologous families at the domain level not only provides more clusters, but also provides more significant signals for developing hypotheses for biological interfaces. Domain-level clusters often have more crystal forms and more sequences than chain-level clusters for proteins in the same family. We provide several examples of both same-Pfam and different-Pfam domain–domain interactions.

Many enzymes have regulatory domains that are involved in dimerization or higher-order oligomerization. ACT domains (Aspartate kinase, Chorismate mutase, TyrA domains) are present in the sequences of many enzymes, and typically bind single amino acids at their dimer interface. The largest cluster of (ACT)/(ACT) domain-level interfaces (Fig. 2a) comprises 14 different multi-domain Pfam architectures, present in 32 crystal forms and 37 PDB entries. At the chain-architecture level, the (ACT)/(ACT) domain dimer is present in a maximum of five crystal forms, demonstrating the power of the domain-level approach. We previously used the (ACT)/(ACT) domain cluster to generate a hypothesis that the ACT domain of human phenylalanine hydroxylase (PAH) would form the (ACT)/(ACT) domain dimer in response to binding of phenylalanine as a mechanism of enzyme activation[11]. While the structure of full-length activated PAH has not been determined, a recent structure of the ACT domain of human PAH with bound Phe contains the ACT dimer present in the ProtCID cluster (PDB: 5FII [https://doi.org/10.2210/pdb5FII/pdb])[29]. Inherited mutations at the domain–domain interface of the PAH ACT domain are associated with phenylketonuria[30]. We can hypothesize that the same ACT dimer is associated with activation in human tryptophan 5-hydroxylases 1 and 2 and tyrosine 3-hydroxylase, which are homologous to PAH and contain similar domain architectures. Human D-3-phosphoglycerate dehydrogenase (PHGDH) also contains an ACT domain that we hypothesize forms the same dimer; the ACT domain in the *E. coli* and *M. tuberculosis* PHGDH proteins are members of the (ACT)/(ACT) domain cluster. This (ACT) domain dimer commonly occurs in other Pfam domains in the same Pfam clan (Supplementary Fig. 1; Supplementary Table 2).

Pfam (tRNA-synt_2d) is the core catalytic domain of phenylalanyl-tRNA synthetases (PheRS). While PheRSs vary significantly in both sequences and structures, the tetrameric quaternary structures are conserved due to the conservation of the catalytic domain[31]. ProtCID contains three tRNA-synt_2d domain-level clusters of interfaces that form in homotetramers and heterotetramers (Fig. 2b). The three clusters contain 13

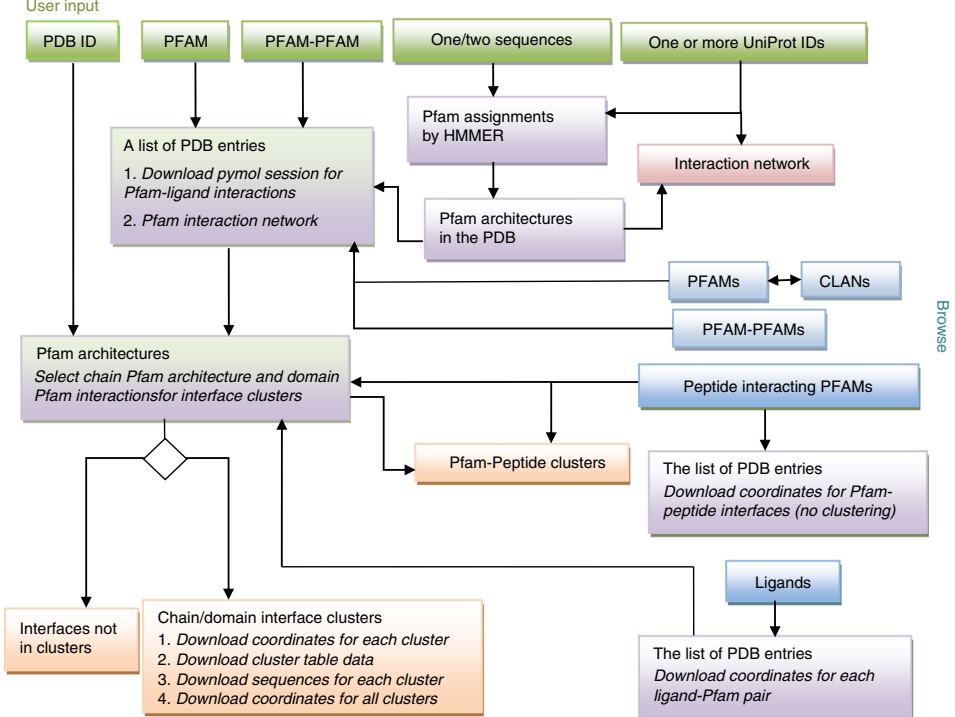

**Fig. 1 The infrastructure of the ProtCID web site.** There are four types of queries that can be input by users: a PDB ID; one or two Pfam IDs; one or two sequences; and one or more UniProt IDs. The user can also browse all available Pfam IDs, Clan IDs, Pfam–Pfam pairs, peptide-interacting Pfams, and ligands. A Pfam page provides a list of PDB structures containing the Pfam. Clicking on one structure leads to the same page as querying by that PDB ID, which shows the Pfam architecture of the entry. From the PDB entry page, a user can select and display interface clusters: chain–chain interfaces, domain–domain interfaces, as well as domain–peptide interfaces. The coordinates of different interactions and clusters can be downloaded from the Pfam pages and the cluster pages.

different chain-architectures (or pairs of architectures) and 16 different Uniprot sequences. Clusters 1, 2, and 3 occur in 16, 9, and 9 different crystal forms respectively. For the heterotetramers (each monomer of which contains a tRNA-synt_2d Pfam domain), clusters 1 and 3 are made of heterodimeric interfaces, and cluster 2 consists of homodimeric interfaces.

Domain clusters between different Pfams can be used to gather structural data on how some domains perform specific binding functions as modules within larger proteins. Ras domain proteins bind proteins from a number of different families, including proteins that contain domains in the (RA) Pfam (Ras-association) family. The human proteome contains 43 different genes that possess (RA) domains. In ProtCID, there is a domain–domain cluster of (Ras) and (RA) that contains structures in 8 crystal forms from 10 PDB entries, 8 Uniprot sequences, and 4 different Pfam chain architectures (Fig. 2c).

In ProtCID, we cluster Pfam domain–domain interactions both between different protein chains and within chains. In some cases, the latter can be used to analyze changes of orientation of two domains within protein chains. For example, adenylation enzymes contain an N-terminal domain (Pfam: AMP-binding) and a smaller C-terminal domain (Pfam: AMP-binding_C) that undergo a relative rotation of 140° after the initial adenylate-forming step[32]. ProtCID has two intra-chain interface clusters of (AMP-binding) and (AMP-binding_C) domains (Fig. 2d). These two clusters have 9 Uniprot sequences in common, and 52 Uniprots in total. In the human proteome, there are 26 proteins that contain both Pfam domains. Only one human protein (ACS2A_HUMAN [https://www.uniprot.org/uniprot/Q08AH3]) has structures in both conformations (PDB: 5IFI [https://doi.org/10.2210/pdb5IFI/pdb] and 5K85 [https://doi.org/10.2210/pdb5K85/pdb]). The two clusters can be used to model these proteins in the two forms.

**Generating hypotheses for oligomeric protein assemblies with ProtCID**. A primary goal of ProtCID is to generate hypotheses of the structures of oligomeric protein assemblies that may not be readily obvious to authors of crystal structures. Many such structures are due to weakly interacting dimers that are facilitated by attachment to the membrane or by scaffolding by other proteins or nucleic acids. To demonstrate the utility of ProtCID, we present several examples of this phenomenon, both confirmed experimentally in the literature and new but provocative hypotheses (Fig. 3).

In 2006, Kuriyan and colleagues discovered the biological relevance of an asymmetric dimer of the EGFR kinase domain[33] that was present in two PDB structures at that time. Extensive experimentation indicated that this interface between the C-terminal domain of one monomer and the N-terminal domain of another monomer served to activate the latter. Later structures of ErbB2[34], ErbB4[35] and a heterodimer of EGFR and ErbB3 were also noted to contain the same dimer[36]. This asymmetric dimer was unexpected because most protein homodimers are symmetric or isologous[2].

The biological effect of this dimer serves as an example of how ProtCID might lead to such a hypothesis. In ProtCID, there is a domain-level cluster of the (Pkinase_Tyr) Pfam that contains the asymmetric dimer from 10 crystal forms and 98 PDB entries of EGFR, one CF and entry of ErbB2, one CF and 3 entries of an ErbB3/EGFR heterodimer, and 2 CFs and 2 entries of ErbB4 (Fig. 3a; Supplementary Fig. 2; a list of ErbB PDB entries and whether they contain the dimer is provided in Supplementary Data 1). This cluster presents a good example of how dimerization may be associated with functional changes. A total of 100 out of 104 structures that contain the asymmetric ErbB dimer contain an active kinase in both positions (91 cases) or

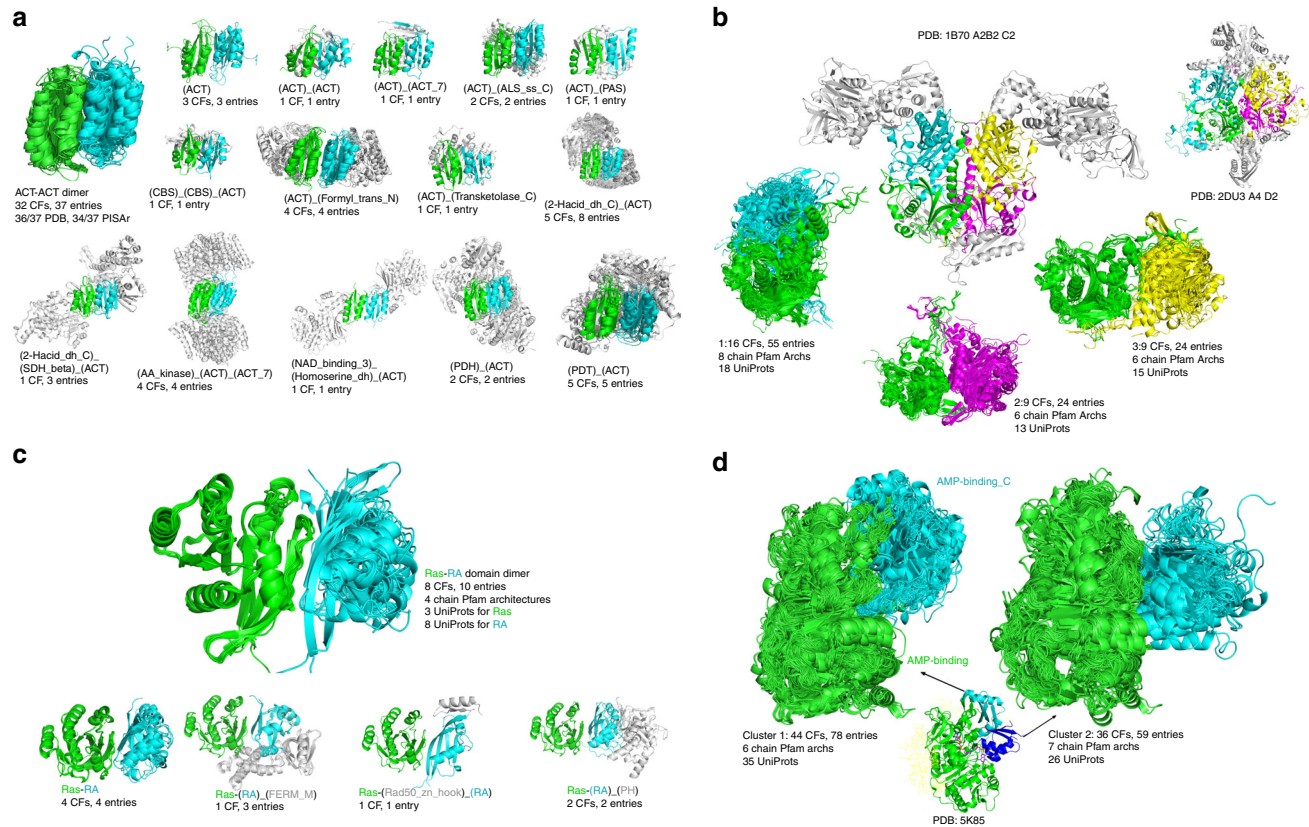

**Fig. 2 Domain clusters. a** (ACT)/(ACT) domain-level cluster containing 14 different chain Pfam architectures. First image shows all domain pairs; each subsequent image shows each chain architecture with the ACT domains in green and cyan. The non-domain segments are colored in light gray. This coloring schema applies to all protein–protein interface figures unless otherwise stated. **b** Three tRNA-synt_2d domain clusters occur in a family of tetrameric tRNA synthetases. The dimer of first cluster is colored in green and cyan, the dimer of cluster 2 is colored in green and purple, and the dimer of cluster 3 is colored in green and yellow. The colors of domains are same in tetramers. One hetero-tetramer (PDB: 1B70, stoichiometry A2B2, symmetry: C2) and one homo-tetramer (PDB: 2DU3 [https://doi.org/10.2210/pdb2DU3/pdb], stoichiometry A4, symmetry: D2) are shown. **c** (Ras)/(RA) is a diff-Pfam domain interface cluster with 8 crystal forms and 10 entries. All Ras domains are single chain domains from three UniProts (RASH_HUMAN, RAP1A_HUMAN and RAP1B_HUMAN). RA domains are in four different chain Pfam architectures from eight UniProts (AB1IP_MOUSE, AFAD_MOUSE, GNDS_RAT, GRB14_HUMAN, KRIT1_HUMAN, PLCE1_HUMAN, RAIN_HUMAN and RASF5_MOUSE). **d** Two intra-chain domain interface clusters of AMP-binding and AMP-binding_C with different domain/domain orientations. These two clusters have nine common UniProts (LUCI_PHOPY, Q8GN86_9BURK, LCFCS_THET8, ACS2A_HUMAN, 4CL2_TOBAC, MENE_BACSU, LGRA_BREPA, J9VFT1_CRYNH, E5XP76_9ACTN) and two common entries (PDB: 5IFI [https://doi.org/10.2210/pdb5IFI/pdb] and 5K85 [https://doi.org/10.2210/pdb5K85/pdb]). These two entries have one UniProt J9VFT1_CRYNH (Acetyl-coenzyme A synthetase from Cryptococcus neoformans), consisting of three monomers each. The chain A and B monomers are in the first conformation and the chain C monomers are in the second conformation. The AMP-binding_C domain of chain C monomer is colored in blue.

only in the activated domain (9 cases). Fifty out of 51 structures of EGFR that do not contain the asymmetric dimer consist only of inactive kinase domains (Supplementary Table 3). As a testament to how difficult it is for biophysical calculations to determine weak biological interactions, PISA predicts the asymmetric dimer of these structures as biological in only 6 of the 98 EGFR entries and all 3 of the heterodimer entries. EPPIC3 identifies the EGFR asymmetric dimer interface as biological from sequence conservation, although because the assembly is asymmetric it does not suggest that the dimer is the biological assembly[37].

The observation of similar interfaces in crystals of homologous proteins can be used to utilize experimental data available on one protein to generate hypotheses for other members of the same family. ProtCID enables this kind of inference in an easily accessible way. One intriguing example is observed for the Pfam domain (Pkinase_Tyr). This Pfam includes tyrosine kinases and most of the tyrosine-kinase-like (TKL) family of kinases. In ProtCID, the TKL kinases BRAF and RAF1 dimers are well represented in the largest cluster for Pfam (Pkinase_Tyr) (Fig. 3b).

The catalytic activity of BRAF and RAF1 is regulated by this side-to-side homodimer of their kinase domains[38,39]. These dimers are relatively weak (kD of ~2 μM)[38] and were initially not recognized[40]; they were first identified by their occurrence in five different crystal forms of BRAF[38].

The cluster also includes five other kinases: TKL-family kinases RIPK2, MLKL, and *Arabidopsis* CTR1, and Tyrosine kinases CSK and ITK (Supplementary Fig. 3a). In ProtCID, this cluster contains 99 structures in 31 crystal forms (a list of PDB entries in the cluster is presented in Supplementary Data 1). The structures of RAF-like kinase CTR1 (PDB: 3PPZ [https://doi.org/10.2210/pdb3PPZ/pdb] and 3P86 [https://www.rcsb.org/structure/3p86]) have been described as similar to the BRAF kinase[41]. Recent analytical ultracentrifugation, mass spectrometry, and mutational data on RIPK2 confirm the significance of RIPK2 in four crystal forms in this ProtCID cluster[42].

The presence of other kinases in the cluster, however, allows us to form hypotheses about their dimerization structures for which there is not yet experimental evidence. Human mixed lineage kinase domain-like protein (MLKL) is a TKL-family

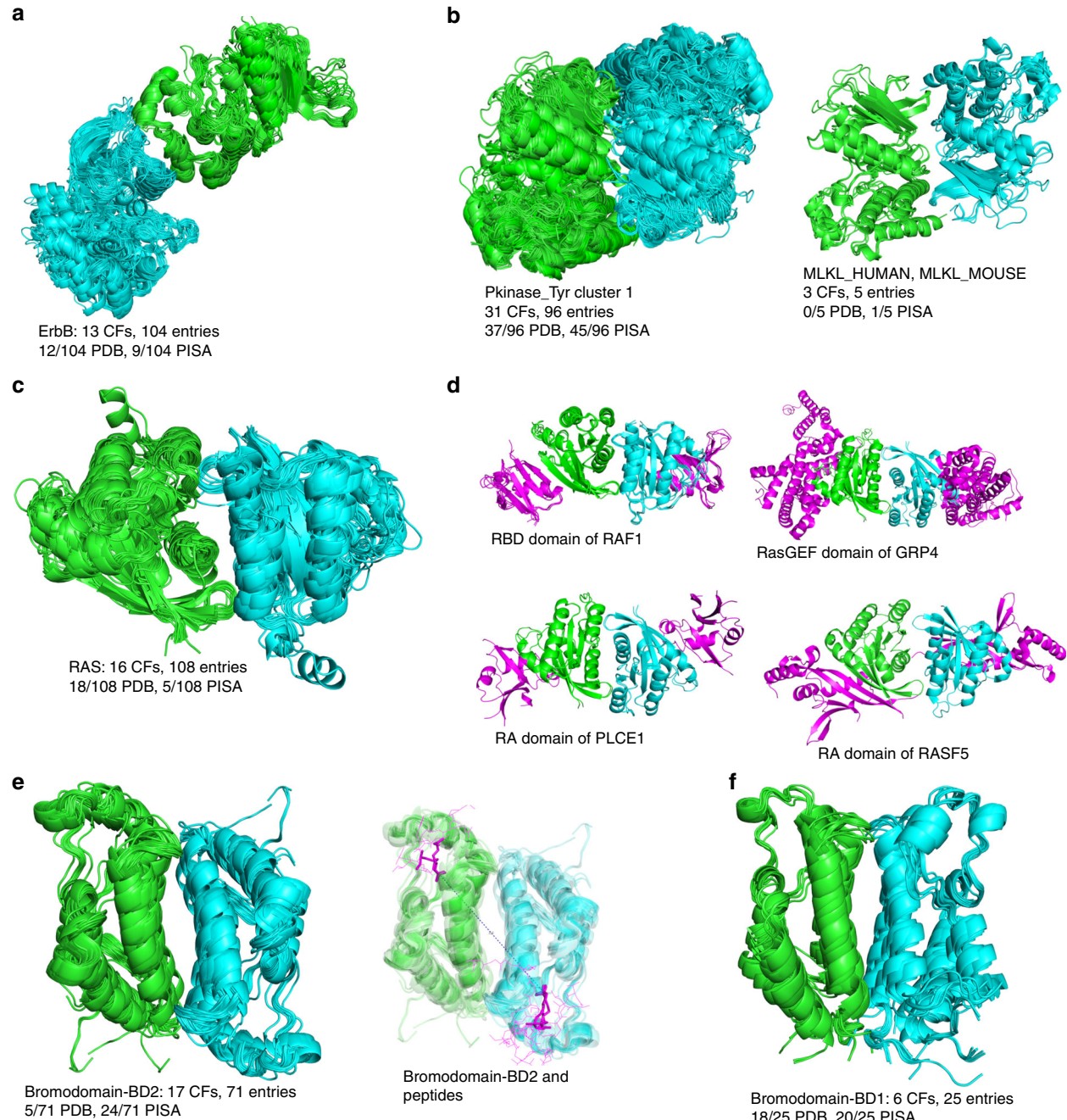

**a** ErbB: 13 CFs, 104 entries
12/104 PDB, 9/104 PISA

**b** Pkinase_Tyr cluster 1
31 CFs, 96 entries
37/96 PDB, 45/96 PISA

MLKL_HUMAN, MLKL_MOUSE
3 CFs, 5 entries
0/5 PDB, 1/5 PISA

**c** RAS: 16 CFs, 108 entries
18/108 PDB, 5/108 PISA

**d** RBD domain of RAF1

RasGEF domain of GRP4

RA domain of PLCE1

RA domain of RASF5

**e** Bromodomain-BD2: 17 CFs, 71 entries
5/71 PDB, 24/71 PISA

Bromodomain-BD2 and peptides

**f** Bromodomain-BD1: 6 CFs, 25 entries
18/25 PDB, 20/25 PISA

**Fig. 3 Hypothesis generation for homodimers. a** The asymmetric dimer of ErbB proteins in a ProtCID cluster of 13 crystal forms (CFs) and 104 PDB entries, comprising homodimers of EGFR, ErbB2, ErbB4, and heterodimers of ErbB3 and EGFR (Supplementary Fig. 2); activating monomer in green; activated monomer in cyan. The PDB biological assembly contains the dimer in 12 of the 104 entries and PISA contains it in 9 of 104 entries. **b** Common domain interfaces in Pkinase_Tyr (PF07714) cluster 1 in ProtCID. The cluster contains 31 CFs and 96 entries. It contains 8 kinases: human BRAF, CSK, ITK, MLKL, RIPK2, and RAF1; mouse MLKL; and *Arapidopsis* CTR1. Inset shows the MLKL dimer. **c** A proposed Ras α4-α5 dimer occurs in 108 crystals structures of human HRAS, rat HRAS, human KRAS, human NRAS, human RAB11B and mouse RND3 (RhoE) (Supplementary Fig. 3). **d** Some crystals of HRAS that contain the α4-α5 dimer also contain Ras binding partners, and thus form heterotetramers. The binding partners are in purple. Top row: the RBD domain of RAF1 kinase (two CFs, PDB: 4G0N [https://doi.org/10.2210/pdb4G0N/pdb] and 4G3X [https://doi.org/10.2210/pdb4G3X/pdb]; 3KUD [https://doi.org/10.2210/pdb3KUD/pdb]); the RasGEF domain of RAS guanyl releasing protein (PDB: 6AXG [https://doi.org/10.2210/pdb6AXG/pdb]). Bottom row: the RA domain of Phospholipase C epsilon (PDB: 2C5L [https://doi.org/10.2210/pdb2C5L/pdb]); and the RA domain of Rassf5 (PDB: 3DDC [https://doi.org/10.2210/pdb3DDC/pdb]). **e** A proposed homodimer found in all structures of the BD2 bromodomain of BET proteins: human BRD2, BRD3, BRD4; mouse BRD4 and BRDT (Supplementary Fig. 4). Inset shows peptide substrates of BD2 H-T homodimers (PDB: 2E3K [https://doi.org/10.2210/pdb2E3K/pdb], 2WP1 [https://doi.org/10.2210/pdb2WP1/pdb], 4KV4 [https://doi.org/10.2210/pdb4KV4/pdb]). The peptides are shown in lines and colored in magenta. The acetyl-lysine (ALY) residues are shown in sticks. The average distance between ALY residues of two peptides is about 30 Å. **f** BET BD1-bromodomain head-to-head (H-H) homodimer. This dimer occurs in 6 CFs and 25 PDB entries and in human BRD2, BRD3, and BRDT and mouse BRDT.

pseudokinase that functions as a substrate of Receptor-interacting serine/threonine kinase 3 (RIPK3) in necroptosis[43]. The cluster shows that MLKL can be dimerized in the same way as BRAF, CRAF, and RIPK2. In fact, all five structures of human MLKL kinase domains in the PDB (in three different crystal forms) contain the BRAF-like dimer. The asymmetric unit of PDB entry 4M69 [https://doi.org/10.2210/pdb4M69/pdb] is a mouse RIPK3-MLKL heterodimer[44]. In this crystal, RIPK3 also forms a BRAF-like dimer. It does not appear in the cluster because RIPK3 is slightly closer to the Pkinase Pfam than the Pkinase_Tyr tetramer. It does appear in a (Pkinase) cluster of 10 CFs and 65 entries. It is therefore possible to build a hypothetical hetero-tetramer of one RIPK3 dimer and one MLKL dimer which are integrated by the RIPK3-MLKL heterodimer interface (Supplementary Fig. 3b). The BRAF-like homodimers of all these structures are not annotated as such in the PDB or described by the authors[44].

HRAS, KRAS, and NRAS form oligomeric structures at the plasma membrane, where they are anchored by palmitolyation and farnesylation[45]. In ProtCID, there is a large domain-based cluster of interfaces for the (Ras) Pfam comprising 16 crystal forms and 108 PDB entries. The cluster includes structures of HRAS (14 CFs, 92 entries), KRAS (3 CFs, 13 entries), NRAS (1 CF, 1 entry), RAB-11B (1 CF, 1 entry), and mouse Rnd3 (1 CF, 1 entry) (Fig. 3c, Supplementary Fig. 4a; a list of HRAS, KRAS, and NRAS PDB entries that do and do not contain the dimer is provided in Supplementary Data 1). The structures are symmetric dimers involving helices α4 and α5 in the interface, with an average surface area of 797 Å$^2$, which is a moderately sized interface for a homodimer. We also find a smaller cluster (5 crystal forms) of a beta dimer, which has been studied by Muratcioglu et al.[46]. By contrast, we do not find the α3–α4 dimer implicated in the same study in any PDB entry.

The α4–α5 dimer in our ProtCID cluster has been implicated as a biologically relevant assembly for NRAS, KRAS, and HRAS. Spencer-Smith et al. found that a nanobody to the α4–α5 surface, as determined by a co-crystal structure of the nanobody with HRAS (PDB: 5E95 [https://doi.org/10.2210/pdb5E95/pdb]), disrupted HRAS nanoclustering and signaling through RAF[47]. Most structures with GTP are considered active, while most structures with GDP are inactive. Of the 92 structures of HRAS in the cluster, 81 of them (88%) are bound with GTP or guanine triphosphate analogs, 9 of them with GDP, and two have no bound ligand (Supplementary Table 4). By contrast, 27 of 54 (50%) HRAS structures that do not contain the α4–α5 dimer are bound with GTP or a triphosphate analog.

Similarly, Ambrogio et al. very recently identified the α4–α5 dimer in their crystal of KRAS[48], and used mutagenesis to show that disruption of this dimer abolished the ability of mutant KRAS to drive tumor growth and the ability of wild-type KRAS to inhibit mutant KRAS. In the 13 KRAS structures in our cluster, 5 are bound with GTP or a triphosphate analog and 8 are GDP bound. Sixty-nine KRAS structures do not contain the dimer. Güldenhaupt et al. performed attenuated total reflection Fourier transform infrared experiments and MD simulations to provide evidence that the α4–α5 dimer may be relevant for NRAS[49]. With ProtCID, we have identified an α4–α5 dimer in a crystal of GNP-bound NRAS (PDB: 5UHV [https://doi.org/10.2210/pdb5UHV/pdb]), that was unrecognized by the authors of this structure[50].

Finally, it has been noted that the α4–α5 Ras homodimer is consistent with binding of Ras effector domains to the surface on the opposite side of the protein from the homodimer interface[51]. Indeed, several of the crystals that contain the α4–α5 dimer also contain Ras binding partners, and therefore consist of (Ras)/Ras-effector heterotetramers, including the (RBD) domain of RAF1

kinase, the (RasGEF) domain of RAS guanyl releasing protein, and the (RA) domains of Phospholipase C epsilon and Rassf5 (Fig. 3d). Other Ras crystals that do not contain the α4–α5 dimer but do contain heterodimeric partners show that the partners could bind to the α4–α5 dimer. These include the (RA) domains of GRB14 and (RALGDS), the (RBD) domain of APOA1, the (PI3Krbd) domain of PK3CG, the (RasGAP) domain of RasGAP, and the (RasGEF) domains of Son-of-Sevenless (Supplementary Fig. 4b).

Bromodomains are modules that bind acetylated lysine residues, primarily in histones[52]. The domain is a four-helix bundle consisting (in sequence order) of αZ, αA, αB, and αC. The largest chain-based cluster contains 17 crystal forms and 71 PDB entries. This head-to-tail symmetric dimer (Fig. 3e) has an interface consisting of the αB and αC helices, with an average surface area of 741 Å$^2$ (a list of PDB entries that contain the dimer is provided in Supplementary Data 1). The Pfam for bromodomains is shorter than the observed domains in structures of these proteins by 28 residues in the C-terminal αC helix. Since the αC helix makes up a substantial portion of the interface in this cluster, the domain-based interfaces fall below our cutoff of 150 Å$^2$ in most of the structures, leaving a related domain-based cluster of only 6 CFs and 32 PDB entries. The distinction highlights the utility of clustering full-length chains as well as Pfam-defined domains to compensate for shortcomings in Pfam's domain definitions.

All of the proteins in this chain-based cluster are members of the BET (Bromodomain and Extra-Terminal domain) family of bromodomain proteins. These proteins, BRD2, BRD3, BRD4, and BRDT, contain two tandem bromodomains, BD1 and BD2, followed by a small extra-terminal (ET) domain towards the C-terminus. In between BD2 and the ET domain there is a coiled-coil Motif B that is associated with homo- and heterodimeriza-tion[53]. The chain-based cluster we have identified consists solely of BD2 domains from BET proteins (Fig. 3e, Supplementary Fig. 5a), including human BRD2, BRD3, and BRD4 as well as mouse BRD4 and BRDT. It is an important observation that every crystal of a BD2 domain from a BET protein in the PDB contains this head-to-tail dimer.

In addition to motif B, the first bromodomain of BET proteins has also been shown to dimerize in solution by Nakamura et al.[54]. The same authors determined the crystal structure of BD1 of human BRD2, and identified a head-to-head symmetric dimer as the likely biological assembly, which was verified by mutation of residues in the interface. The BD1 head-to-head cluster is the fifth largest chain-based cluster of bromodomains, comprising six crystal forms and 25 PDB entries (Fig. 3f, Supplementary Fig. 5b), including BD1 of human BRD2 and BRD3 and both mouse and human BRDT. The head-to-head dimer cluster contains only 25 of 235 PDB entries with BD1 domains of BET proteins.

While there is strong evidence for dimerization of full-length BET proteins, there is not yet specific evidence that the BD2 domains homodimerize in vitro or in vivo. Huang et al found that the BD2 of BRD2 expressed as a single-domain protein did not form stable dimers in solution[55]. However, there is a possibility that if the BD1 domains and motif B of a BET protein homodimerize, then the BD2 domain that sits in between them in the sequence may form homodimers given the increased concentration induced by the dimerization of the flanking domains. The dimer we have identified with ProtCID is a strong candidate for the form of a BD2 dimer, if it exists, and serves as an example of how ProtCID is able to generate credible and testable hypotheses for the formation of weak or transient interactions of proteins.

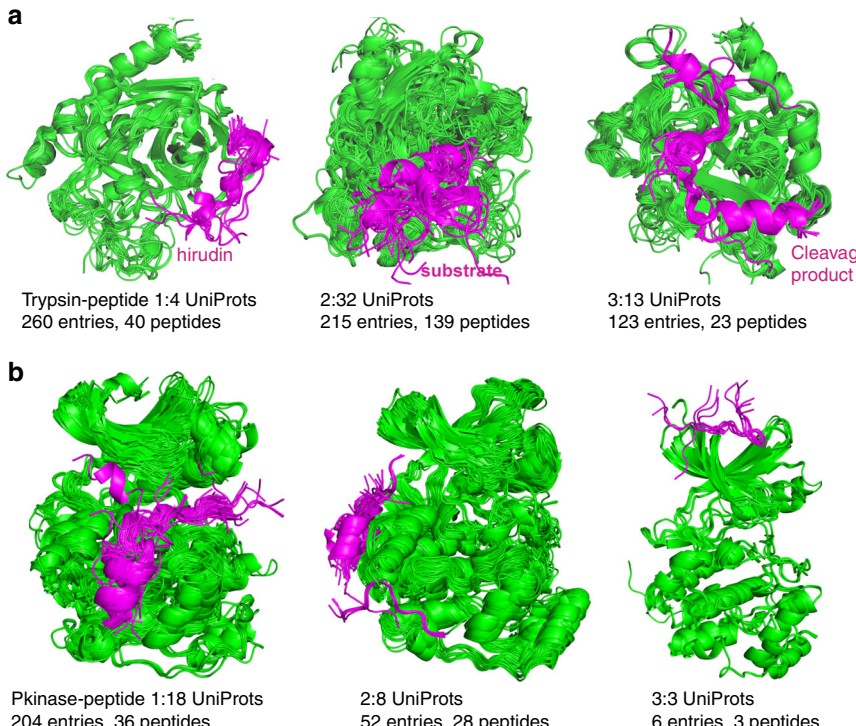

**Fig. 4 Pfam-peptide interactions. a** Three trypsin-peptide interface clusters. The number of peptides indicates the number of unique peptide sequences; the peptide sequences of each cluster can be found on each cluster page on ProtCID. The Pfam domains are colored in green and the peptides are colored in magenta. The first cluster mostly constists of human thrombin bound to peptides from hirudin from leeches. The peptides of the second cluster is a large set of substrates bound to the active sites of trypsin-like proteases. The peptides of the third cluster are cleavage products, created by internal cleavage by activating enzymes. **b** Pkinase–peptide interactions show the binding sites of substrates, activators, and inhibitors. Cluster 1 shows the peptides binding to active sites as inhibitors or substrates. Cluster 2 contains peptides binding to a groove between $\alpha_d$ and $\alpha_e$ and the β7–β8 reverse turn on the kinase C-terminal domain. Cluster 3 is a small cluster of peptides binding to the N-terminal groove.

**Peptide-binding domains in ProtCID**. The function of many protein domains is to bind peptides from other proteins. We define a peptide as a protein chain with less than 30 residues. ProtCID provides data on 1083 Pfam-domains with peptide interactions in the PDB. In ProtCID, we cluster the structures of peptides bound to Pfam domains, so that each cluster provides structural information on peptide–domain interactions with specific functions. Some protein families bind peptides at multiple sites with different functional attributes.

As an example, Trypsin domains have three large peptide-interface clusters in ProtCID (Fig. 4a). The first is a set of 260 entries mostly consisting of human thrombin bound to a peptide from hirudin from leeches[56]. The cluster also includes a few structures of complexes of thrombin with mouse or human proteinase activated receptor 1 and 3 (PAR1, PAR3), which mimic hirudins[57]. The second large cluster is a large set of inhibitor peptides bound to the active sites of more than a dozen trypsin-like protease family members. The third cluster is entirely made of complexes of the heavy and light chains of some trypsin-like proteases, including thrombin, chymotrypsin, urokinase, acrosin, and matriptase. These complexes are created by internal cleavage by activating enzymes.

Serine/Threonine kinases (Pkinase) have three peptide-interface clusters in ProtCID (Fig. 4b). The first contains 36 distinct peptides binding to the kinase active site. While 159 out of 204 of these structures consist of interactions between kinase domains and peptides from small inhibitor proteins such as PKI-alpha, the remainder are structures of kinases with substrate or substrate analog peptides. The second Pkinase/peptide cluster comprises 28 different peptides binding to a docking groove on

the C-terminal domain of 21 different kinases. The peptide-binding induces conformational changes in the active site[58]. The last cluster includes fragments of the beta subunit of casein kinase bound to the N-terminal domain of the alpha subunit of casein kinases.

The human proteome contains many protein domain families that primarily function as peptide-binding modules, often within larger proteins that also contain other domains. If two proteins can be demonstrated to interact, for instance through high-throughput protein–protein interaction studies, and one of them contains a peptide-binding domain, then in many cases a reasonable hypothesis is that the peptide-binding domain of one protein may bind to an intrinsically disordered region of the other. We have therefore used ProtCID to compile a list of the more common peptide-binding domain families within the human proteome with available structural information in the PDB. We include only domains that primarily function as non-enzymatic peptide binders and are well represented in the PDB and the human proteome (see Methods section). We refer to these as Professional peptide binding domains (PPBDs). There are currently a total of 42 PPBD Pfam families with peptide-bound structures in the PDB (Supplementary Table 5). These domains are present in 1051 human protein sequences, or about 5% of the proteome. Identifying these domains can be useful in analyzing protein-protein interaction networks (see below).

**Ligand and nucleic-acid/protein interactions in ProtCID**. We define all non-polymer molecules except water in the PDB as ligands. In ProtCID, the ligands are clustered based on the extent

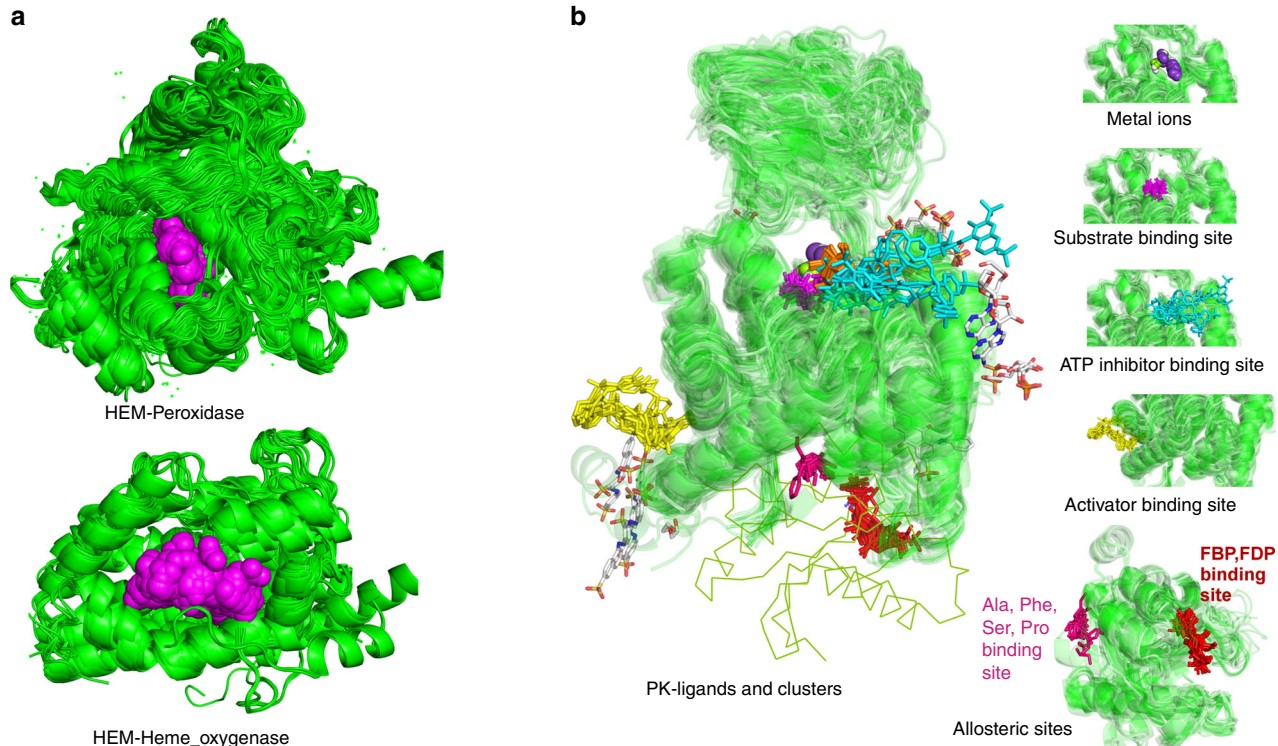

**Fig. 5 Pfam–ligand interactions. a** The interactions of Heme (PDB: HEM[http://www.rcsb.org/ligand/HEM]) and Pfams. The coordinates of HEM–Pfams interactions can be downloaded from the ligand web page in file HEM.tar.gz file. The file provides the domain coordinates of all Pfams that interact with heme (HEM) ligand and the PyMOL scripts for each Pfam. HEM/Peroxidase interactions (top) and HEM/Heme_oxygenase interactions (bottom) are generated by peroxidase_HEM_pairFitDomain.pml and Heme_oxygenase_HEM_pairFitDomain.pml respectively. Hemes are shown in spheres and colored in magenta. **b** The interaction clusters of Pyruvate Kinase (PK) and ligands are generated from PK_pdb.tar.gz file which can be downloaded from PK web page. The ligands and clusters are represented as selection objects in PyMOL. Different ligand clusters are shown in different colors. The C-terminal domain (PK_C) is added in lines to show the full-length pyruvate kinase.

to which they share Pfam HMM positions that they contact. ProtCID provides analysis and coordinates for two different views of Pfam–ligands interactions: (1) one ligand and its interacting Pfams; (2) one Pfam and its interacting ligands. Figure 5a displays the interactions of heme (PDB identifier HEM) binding to two different Pfams, Peroxidase domains and Heme oxygenase domains. Heme binds to 134 different Pfams in 4319 PDB entries. Figure 5b shows the major clusters of the interactions of pyruvate kinases (PK) and their ligands, including sites for metal ions, substrates, allosteric activators such as fructose-1,6-bisphosphate (FBP), and allosteric inhibitors such as ATP and alanine. Pfam–ligand interactions can be queried in ProtCID by Pfam ID, or browsing Pfams and ligands in the Browse page (Supplementary Table 6 and Supplementary Data 2).

In ProtCID, nucleic acids are treated in the same way as other ligands. This makes it very easy to find all structures of proteins from a nucleic-acid binding domain family with bound DNA or RNA. A total of 1260 Pfams interact with DNA or RNA in 6,504 PDB entries (Supplementary Table 7, Supplementary Data 2). We provide several examples (Fig. 6). The Pfam (HLH) is the basic helix-loop-helix DNA-binding domain characterized by two α-helices connected by a loop. In the PDB, there are 22 structures of (HLH) domains bound to double-stranded DNA (Fig. 6a) from 22 distinct UniProts including 11 human proteins. The human proteome contains 115 UniProts with (HLH) domains. The Pfam (dsrm) is a double-stranded RNA binding domain. There are 33 entries and 11 UniProts containing the (dsrm)-dsRNA interaction in the PDB (Fig. 6b). The human proteome contains 20 proteins with this domain, only two of which (TRBP2_HUMAN

[https://www.uniprot.org/uniprot/Q15633] and DHX9_HUMAN [https://www.uniprot.org/uniprot/Q08211]) have structures in the PDB with RNA. (DEAD) domains are a family of helicases that unwind nucleic acids, including both DNA and RNA. The interaction of (DEAD) domains and single-stranded DNA occurs in 19 entries and 6 UniProts in the PDB (Fig. 6c). The interaction of (DEAD) domains and RNA occurs in 61 PDB entries and 24 UniProts (Fig. 6d). The (DEAD) domains bind to ssDNA and ssRNA in similar ways. Currently, the PDB has structures of 94 UniProts containing (DEAD) domains. The mode of binding to ssDNA or ssRNA of these additional proteins can be modeled using the known DEAD-domain/nucleic-acid structures in the ProtCID cluster.

**Structural data for protein–protein interaction networks.** Many proteins are part of large protein complexes or molecular machines with many different components and thus interact both directly with some partners and indirectly with many others. Large-scale studies of protein–protein interactions have also identified hub proteins that participate in a large number of interactions with other proteins[59–61]. We have enabled a search function in ProtCID which takes a list of Uniprot identifiers and searches for structural information on interactions between the proteins in the list. It performs this task by identifying the Pfams in each of the queries, and then searching for ProtCID domain–domain data for those Pfams. It also identifies peptide binding domains in the queries. It can be run in two modes: (1) all-against-all for the structural analysis of large protein

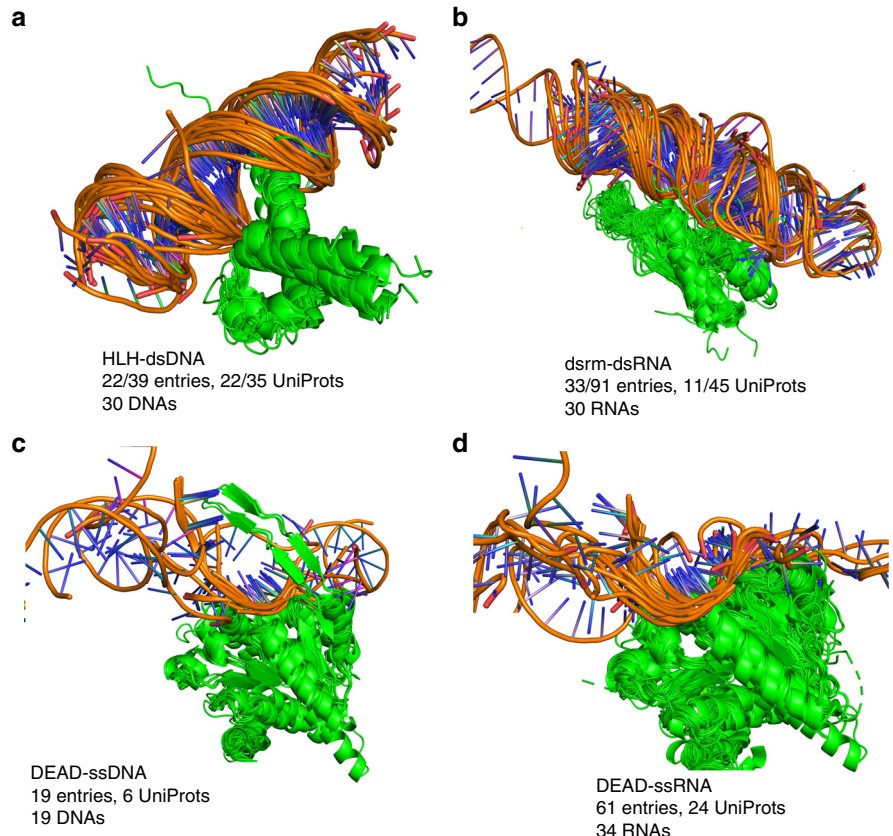

**a**

HLH-dsDNA
22/39 entries, 22/35 UniProts
30 DNAs

**b**

dsrm-dsRNA
33/91 entries, 11/45 UniProts
30 RNAs

**c**

DEAD-ssDNA
19 entries, 6 UniProts
19 DNAs

**d**

DEAD-ssRNA
61 entries, 24 UniProts
34 RNAs

**Fig. 6 Pfam–nucleic acids interactions. a** The interaction of Pfam HLH domain and double-stranded DNAs contains 22 of 39 entries with HLH domains in the PDB and 22 of 35 UniProts with 30 unique DNA sequences. HLH domains are aligned based on Pfam hidden Markov model positions by pair_fit in PyMOL. The coordinates and PyMOL script files can be downloaded from http://dunbrack2.fccc.edu/ProtCiD/Results/EntityPfamArchWithPfam.aspx? PfamId=HLH web page and are included in HLH_pdb.tar.gz with other ligands or http://dunbrack2.fccc.edu/ProtCiD/IPdbfam/PfamLigands.aspx? Ligand=DNA web page and are included in DNA.tar.gz with other DNA–interacting Pfams. **b**. The interaction of Pfam dsrm domains and double-stranded RNAs occurs in 33 of 91 entries with the (dsrm) Pfam, and 11 of 45 UniProts. **c**. The interaction of Pfam DEAD domains and single-stranded DNAs contains 19 entries and 6 UniProts. **d** The interaction of Pfam DEAD domains and single-stranded RNAs occurs in 61 entries and 24 UniProts, interacting with 34 distinct RNA sequences. The DEAD domains of D3TI84_ANAPL [https://www.uniprot.org/uniprot/D3TI84] (PDB: 4A36 [https://doi.org/10.2210/pdb4A36/pdb]) and DDX58_HUMAN [https://www.uniprot.org/uniprot/O95786] (9 entries) structures interact with dsRNA (not shown). Nucleic acids are colored in orange.

complexes; (2) one-against-all for the structural analysis of hub proteins.

As an example, the human HBO1 complex is a histone H4-specific acetyltransferase composed of four proteins: ING4, EAF6, JADE3, and KAT7. The p53 pathway is a main target of the complex[62]. The ProtCID Uniprot search of these four proteins returns an interaction network based on four Pfam domain-domain clusters (Fig. 7a). They are derived from four structures of the Yeast HBO complex (Fig. 7b), containing four proteins, EAF4, EAF6, EPL1, and ESA1, which contain (ING) (NuA4), (EPL1), and (MOZ_SAS) domains, respectively, as do the four human proteins (ING4, EAF6, JADE3, and KAT7 also, respectively). All four domain clusters contains 2 CFs and 4 entries. In the PDB, there is one structure (PDB: 5GK9 [https://doi.org/10.2210/pdb5GK9/pdb]) containing the (MOZ_SAS) domain of KAT7_HUMAN[https://www.uniprot.org/uniprot/O95251], and one structure (PDB: 4AFL [https://doi.org/10.2210/pdb4AFL/pdb]) containing the (ING) domain of ING4_HUMAN[https://www.uniprot.org/uniprot/Q9UNL4]. We built homology models of JADE3_HUMAN [https://www.uniprot.org/uniprot/Q92613] and EAF6_HUMAN [https://www.uniprot.org/uniprot/Q9HAF1] with SwissModel[63] using the yeast structures as templates, and then superposed these models, and the human proteins in PDB entries 5GK9 and

4AFLonto the homologous chains in the yeast HBO complex (PDB: 5J9Q [https://doi.org/10.2210/pdb5J9Q/pdb]) to build a model of the human HBO1 complex (Fig. 7c). This example demonstrates the utility of ProtCID in identifying the structures of complexes and individual proteins in the PDB that can be used to model the three-dimensional structures of target complexes of interest.

The one-against-all mode of the Uniprot protein search page in ProtCID returns a list of potential domain–domain and PPBD–peptide interactions that may explain how the hub protein interacts with each of its partners. Figure 7d shows the interactions in the PDB between P53 and its interactors based on Pfams and PPBDs. ProtCID identifies PPBDs (Supplementary Table 5) in both hub and partner proteins so that this mode of binding is also presented to the user as a viable hypothesis. For example, studies show that a peptide segment of the p53 C-terminus binds to 14-3-3 proteins[64]; however there is no structure of this interaction in the PDB. A potential interaction between a peptide of P53 and the PPBD 14-3-3 proteins is identified by an edge in the network in Fig. 7d.

**Modeling interactions for Pfams in clans.** We have added access to ProtCID Pfam-based clusters at the Clan (superfamily) level. This is enabled through a page for each clan (e.g., http://dunbrack2.fccc.

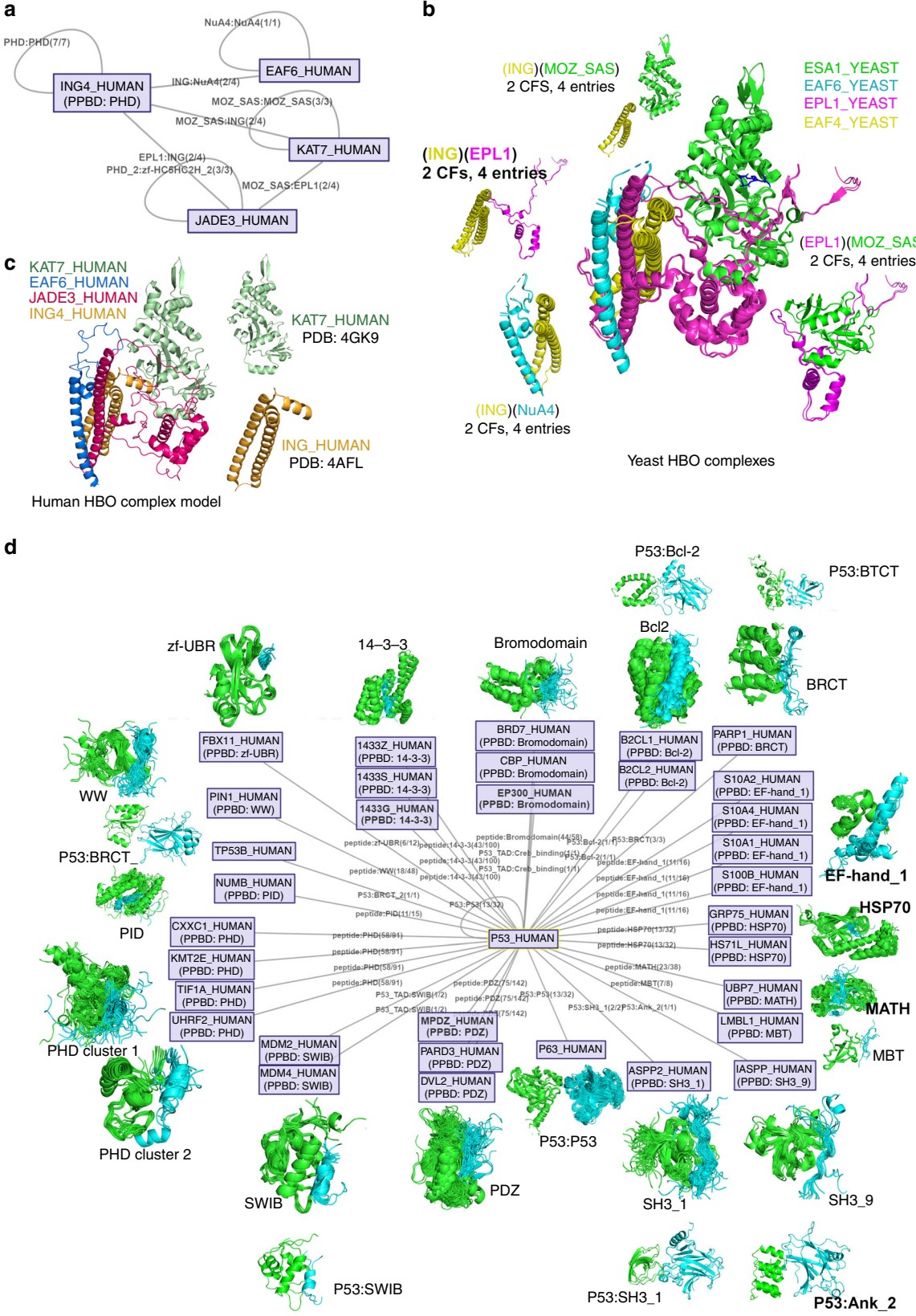

edu/ProtCiD/Results/ClanPfams.aspx?ClanID=ATP-grasp), which lists Pfams that have structures with nucleic acids or peptide and Pfams do not. The numbers of Pfams, UniProts, and human UniProts for a clan and whether they interact with peptides and nucleic acids in the PDB are provided in Supplementary Table 8

and Supplementary Data 2. The information on which Pfams in a clan are represented by structures in the PDB with bound nucleic acids or peptides is particularly useful in modeling.

For example, there are 14 Pfam families in the clan SH3, 11 of which are represented in structures in the PDB. However, only

**Fig. 7 Hypotheses for the structures of biological complexes and hub protein complexes. a** Cytoscape interaction network of human HBO complex connected by four domain clusters. HBO is a histone acetyltransferase binding to origin recognition complex subunit 1 (ORC1). UniProts include EAF6_HUMAN [https://www.uniprot.org/uniprot/Q9HAF1], ING4_HUMAN [https://www.uniprot.org/uniprot/Q9UNL4], JADE3_HUMAN [https://www.uniprot.org/uniprot/Q92613] and KAT7_HUMAN [https://www.uniprot.org/uniprot/O95251]. The edge label shows the Pfam pair and the number of crystal forms and the number of entries in the largest cluster. **b** Yeast HBO complexes and domain clusters. A yeast complex is composed of EAF6_YEAST [https://www.uniprot.org/uniprot/P47128] (cyan), EPL1_YEAST [https://www.uniprot.org/uniprot/P43572] (magenta), ESA1_YEAST [https://www.uniprot.org/uniprot/Q08649] (green), and YNG2_YEAST [https://www.uniprot.org/uniprot/P38806] (yellow), containing (NuA4), (EPL1), (MOZ_SAS), and (ING) Pfam domains respectively. The complex is connected by 4 ProtCID domain clusters, each of which contains 2 crystal forms and 4 entries. **c** Model of the human HBO1 complex constructed by superposing the experimental structures of KAT7_HUMAN and ING4_HUMAN and predicted structures of EAF6_HUMAN and JADE3_HUMAN onto the yeast HBO1 structure. The model of JADE3_HUMAN was built by removing the sequence segments of PHD_2 and zf-HC5HC2H_2 domains for better sequence alignment. **d** The interactions of hub protein P53_HUMAN. PPBD indicates a professional peptide binding domain. An edge is labeled by Pfam:Pfam if there are crystal structures in PDB, or by peptide:Pfam if the interactor contains a PPBD. If there are no PDB entries containing the interfaces of two UniProt nodes, but one of them has a PPBD, an interaction is predicted and a peptide: PPBD edge is added. For peptide:PPBD, the number of crystal forms and the number of entries are counted from the peptide interface cluster. Users click an edge to retrieve the domain interactions between two node proteins and their clusters. Each node is also clickable to query the structures of the node protein. The Pfam assignments and the complete list of interactions for the input UniProts are provided in table format by clicking the links above the Cytoscape picture.

five of these Pfam families in the clan are represented by structures of SH3 domains with bound peptides, three of which are shown in Fig. 8a (top row). These domain-peptide clusters indicate that Pfams in the SH3 clan typically bind peptides in the same way. This observation can be used to derive hypotheses of how structures in other Pfams in the same clan with no available peptide-bound structures may bind peptides. Three examples are shown in Fig. 8a (bottom row).

Similarly, we can use ProtCID to develop structural hypotheses for proteins in nucleic-acid binding Pfams that do not yet have nucleic-acid bound structures in the PDB. For example, the clan page for DSRM lists the 7 Pfams in this clan, including (dsrm) shown in Fig. 6b. Six of these Pfams are in the PDB but only three of them contain bound RNA and one additional Pfam contains bound DNA. Structures of the two remaining Pfams in the PDB with no nucleic acids, (DND1_DSRM) and (Staufen_C), can be superposed on (dsrm) structures to produce models of how they might bind to RNA (Fig. 8b). The (DND1)_(DSRM) domain is contained in human proteins A1CF_HUMAN[https://www.uniprot.org/uniprot/Q9NQ94] (APOBEC1 complementation factor) and DND1_HUMAN[https://www.uniprot.org/uniprot/Q8IYX4] (Dead end protein homolog 1), among others. Pfam (Staufen_C) is found in human proteins STAU1, STAU2, PRKRA, and TARBP2.

## Discussion

The Protein Data Bank holds a tremendous amount of information on the interactions of molecules in biological systems, but it can be difficult to access and analyze rapidly. ProtCID is designed to accomplish this task with a unique combination of features that enable users to obtain the rich structural information available for any biological system quickly and easily. Some of these features are individually or jointly available in other databases but all of them are necessary to access the full range of structural information currently available. We discuss several of these features in turn.

First, for crystallographic structures, we build the coordinates of set of 27 unit cells (3 × 3 × 3) and determine the entire set of unique protein–protein interfaces present within the crystal. These form the basis of all further analysis in ProtCID. Many databases that provide data on protein–protein interactions only analyze coordinates within the asymmetric unit and/or within the biological assemblies deposited in the PDB, missing interactions that are not annotated in the PDB.

Second, the clustering of interactions is crucial for providing evidence in favor of the biological relevance of any specific

interaction analyzed in ProtCID. Some protein families have evolved different ways of forming oligomers in different branches of their phylogenetic trees. These interaction will appear in distinct ProtCID clusters with non-overlapping protein sets. In other cases, some proteins form larger oligomers that contain two or more distinct interfaces. Some proteins exist in different oligomers under different conditions with different functional properties[65]. ProtCID helps to sort out which PDB entries contain which interfaces of these oligomers. By contrast, some databases lump all interactions of domains into one group, and do not distinguish between different modes of binding.

Third, for chain–chain and domain–domain interaction clusters, ProtCID provides the number of crystal forms. Larger numbers of crystal forms and higher percentages of all available crystal forms for an interaction type indicate that the interfaces in the crystal are likely to be biological[4]. As far as we are aware, providing the number of crystal forms is a unique feature of ProtCID among protein–protein interaction databases and servers.

Fourth, for all clustered protein interactions, ProtCID provides a link to download all the coordinates in the cluster and scripts to visualize the interactions. Many webservers provide the coordinates of each structure for some type of interaction but downloading each structure requires a click. There can be 100s of structures for an interaction of interest. Some allow online visualization but no coordinate download; some do not provide coordinates at all. Without the coordinates, it is not possible to perform any further analysis of the complexes.

Finally, ProtCID provides complete annotations of the members of each cluster for protein-protein interactions (at the chain and domain levels) and for interactions of domains with peptides, ligands, and nucleic acids. These annotations include: PDBid and chain, crystal form, UniProt identifiers (e.g., BRCA1_HUMAN [https://www.uniprot.org/uniprot/P38398] instead of P38398), species, Pfam domain and chain architectures (e.g., (Ras) instead of PF00071), and whether interactions are present in the biological assemblies deposited by authors and calculated by PISA. The last of these provides information on whether an interface is regarded as biological by the authors of structures, which then may be followed up in the papers describing the structures. Many online databases do not contain information on the identities of proteins in the complexes they list, instead providing only PDB identifiers and links to the PDB's website. Clicking through 100s of structures this way is not viable.

ProtCID has some limitations based on the data that are currently available in the PDB. For some protein families, there may

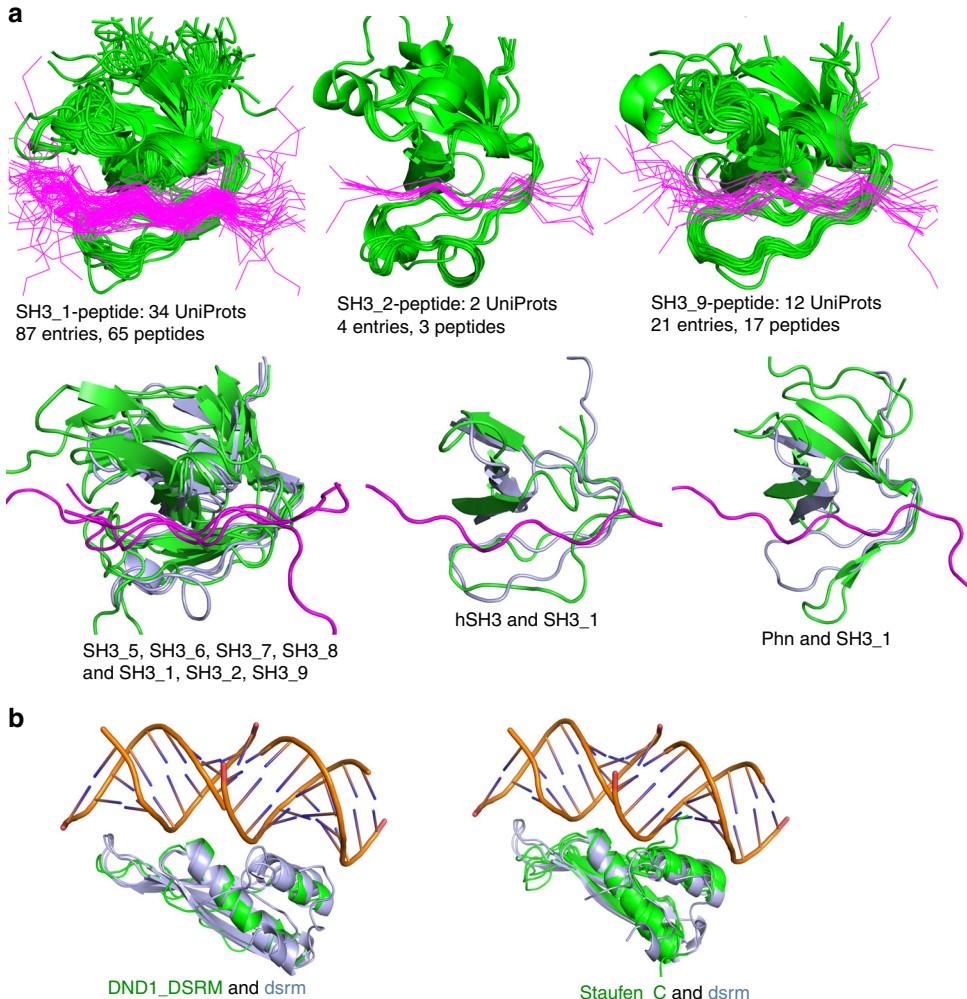

SH3_1-peptide: 34 UniProts
87 entries, 65 peptides

SH3_2-peptide: 2 UniProts
4 entries, 3 peptides

SH3_9-peptide: 12 UniProts
21 entries, 17 peptides

SH3_5, SH3_6, SH3_7, SH3_8
and SH3_1, SH3_2, SH3_9

hSH3 and SH3_1

Phn and SH3_1

DND1_DSRM and dsrm

Staufen_C and dsrm

**Fig. 8 Peptide-binding and DNA-binding to Pfams in clans. a** Models of peptide binding to Pfams in the SH3 clan. First row: SH3_1/peptide interface cluster, SH3_2/peptide cluster and SH3_9/peptide cluster. These clusters in the SH3 clan show similar peptide-binding grooves in different Pfams. Peptides are shown in lines and colored in purple. Second row: the same peptide binding groove after superpositions of the structures of SH3_5, SH3_6, SH3_7, SH3_8, hSH3 and Phn (colored in green) onto the SH3_1, SH3_2 and SH3_9 (colored in light blue) peptide-bound structures. **b** Modeling Pfam-RNA binding in the DSRM clan of double-strand RNA binding proteins. The DSRM clan contains 7 Pfams, of which 4 Pfams (Dicer_dimer, dsrm, Rad52_Rad22 and Ribosomal_S5) have RNA-bound structures in the PDB. Pfams DND1_DSRM (1 UniProt: DCL1_ARATH [https://www.uniprot.org/uniprot/Q9SP32], 1 PDB entry: 2LRS [https://doi.org/10.2210/pdb2LRS/pdb]) and Staufen_C (4 UniProts: STAU1_HUMAN [https://www.uniprot.org/uniprot/O95793], TRBP2_HUMAN [https://www.uniprot.org/uniprot/Q15633], Q9VJY9_DROME [https://www.uniprot.org/uniprot/Q9VJY9], and STAU_DROME [https://www.uniprot.org/uniprot/P25159], 4 entries: 4DKK [https://doi.org/10.2210/pdb4DKK/pdb], 4WYQ [https://doi.org/10.2210/pdb4WYQ/pdb], 4X8W [https://doi.org/10.2210/pdb4X8W/pdb], and 5CFF [https://doi.org/10.2210/pdb5CFF/pdb]) do not have RNA-bound structures in the PDB. Both DND1_DSRM and Staufen_C domains (colored in green) superpose well to (dsrm) domains (colored in light blue). Pfam LIX1 does not have structures in the PDB.

be just one crystal form (even if there are multiple entries), and thus there will be no clusterable interfaces across crystal forms. ProtCID depends on Pfam, and there are some proteins that are poorly annotated by Pfam. Pfam misses some N or C terminal elements of secondary structure that may form a large portion of a homodimer or heterodimer interface. This compromises our ability to cluster some domain–domain interfaces. We compensate for this in part by continuing to provide the chain–chain clustering, since these include the entirety of each chain.

In sum, we hope that with the extension of ProtCID to protein domains and further enhancements described in this paper, it will prove useful in helping users to access the vast amount of structural information that can be used to understand the properties of biological systems.

## Methods

**ProtCID databases**. To create ProtCID, we first assign Pfam domains to all unique sequences in the Protein Data Bank with a procedure we developed for this purpose[7]. Our protocol employs alignment of hidden Markov models (HMMs) of PDB protein sequences to Pfam HMMs as well as structure alignments for validating the assignments of weakly scoring hits. We carefully annotate split domains, which are protein domains with other domains inserted within them, and correctly identify a larger number of repeats in repeat-containing proteins than Pfam does. With these Pfam assignments, we define a Pfam architecture for each protein in the PDB, which is defined as the ordered list of Pfams along the chain along with their positions in the sequence. For example, a protein with one kinase domain and one SH2 domain in that order is annotated as (Pkinase)_(SH2). Supplementary Fig. 6 shows the procedure we use to generate the interface clusters. In Supplementary Fig. 6, the Sulfotransferase cluster contains 70 entries in 37 crystal forms.

Entries are assigned to the same crystal form if: (1) they have the same entry Pfam architectures; (2) the same space group; (3) the same asymmetric unit stoichiometry of protein chains; (4) crystal cell dimensions and angles within 1%

variance. We then compare the interfaces of crystals with the same Pfam entry architectures and if at least two thirds of their interfaces are highly similar interfaces, then we merge the crystal forms into one crystal form. This can happen when two different structures are essentially the same crystal and contain all of the same interfaces, but one is solved as an asymmetric unit monomer and a space group with $N$ symmetry operators and the other is solved as an asymmetric unit dimer and a related space group with $N/2$ symmetry operators.

Chain interfaces are generated from a collection of 27 unit cells arranged in a $3 \times 3 \times 3$ lattice. The domain interfaces are generated from chain interfaces by the Pfam-defined start and stop positions within the full-length chains. Two domains (or chains) are considered to be interacting if and only if they have at least ten pairs of $C_\beta$ atoms with distance ≤12 Å and at least one atomic contact ≤5 Å, or at least five atomic contacts ≤5 Å.

To measure the similarity of two interfaces, we must have a correspondence of the residues in one interface with homologous positions in the other, especially when the proteins are homologous and not identical sequences. Since we have an alignment of every structure to Pfams, we can use the Pfam HMM positions to identify homologous positions in two homologous proteins or protein domains (as long as the HMM covers 80% of the shorter domain). The similarity of interface pairs was calculated with a Jaccard-index[66] metric, $Q$ described by Xu et al.[4], which is equal to a weighted count of the common contacting residue pairs in two interfaces divided by the total number of unique pairs in the two interfaces. A value of $Q$ of 1 means two interfaces are interacting in an identical way. A value of $Q$ of 0 means there are no common contacts. We cluster chain and domain interfaces with surface area >100 Å$^2$ by a hierarchical average-linkage clustering algorithm. Initially each interface is initialized to be in its own cluster. At each step, the two clusters with the highest average $Q$-score between them are merged as long as their $Q_{avg} \geq 0.30$. When no two clusters can be merged with $Q_{avg} \geq 0.30$, then the algorithm is stopped.

We cluster interfaces of all pairs of individual Pfam domains from single and multi-domain proteins. This includes both same-Pfam pairs and different Pfam pairs. Same-Pfam pairs are mostly homodimers, but they also include some interchain homologous heterodimers and intrachain domain–domain interactions with the same Pfam. Different-Pfam pairs are heterodimers (domains with different sequences from two different protein chains) but they also include some intrachain domain–domain interactions. For example, we cluster all (Pkinase): (Pkinase), (SH2):(SH2), and (Pkinase):(SH2) interfaces, regardless of what single-domain or multi-domain architectures these domains come from. For each homodimeric or heterodimeric pair of Pfam chain architectures or Pfam domains, ProtCID reports clusters that contain the interface (sorted by the number of crystal forms they are observed in), as well as the PDB entries that contain them and the interface surface areas, and whether or not the interface is present in the author or PISA-defined biological assemblies for these entries. Links are provided to download the coordinates of the domain and chain dimers and PyMol scripts to visualize them.

**Pfam–Peptide interfaces**. Interactions with peptides are defined only for Pfam domains, not entire chains. This is because most peptide-binding domains are modular and are reused in different contexts with other domain types. A peptide is defined as any protein chain with length less than 30 amino acids in the PDB. The chain type is based on the attribute "Polymer Type" defined in the PDB mmCIF files; a protein or peptide chain has the Polymer Type defined as polypeptide. A domain–peptide interface is defined as an interaction with ≥10 $C_\beta$–$C_\beta$ contacts with distance ≤12 Å, or ≥5 atomic contacts with distance ≤5 Å. If a peptide contacts several chains in a biological assembly, the interface with ≥75% atomic contacts is used as the peptide-interacting interface; otherwise, we keep all the interfaces. For any two Pfam-peptide interfaces in the same Pfam, the number of same-Pfam HMM positions interacting with peptides are counted as $N_{hmm}$. We use the pair_fit command in PyMOL to superpose coordinates of the domain interfaces via their common Pfam HMM positions. We calculate the minimum RMSD ($RMSD_{pep}$) between the peptides by linear least-squares fit of all possible sequence alignments between them. We cluster Pfam–peptide interfaces using a hierarchical average-linkage clustering algorithm by $N_{hmm}$ and $RMSD_{pep}$. In this method, each interface is initialized to be a cluster. At each step, the two clusters with the largest number of common interacting Pfam HMM sites are merged, as long as the $N_{hmm} \geq 3$ and $RMSD_{pep} \leq 10$ Å.

We used several criteria to define professional peptide binding domains (PPBDs):

1. The primary function of the domain must be peptide-binding in most proteins that contain the domain. Some domains primarily perform other functions such as binding DNA or other folded protein domains, and their functions are modified by peptide binding; these are excluded from PPBDs. In most cases, there is a common motif, often confined to one amino acid position, that demonstrates that peptide binding was a function of the common ancestor of proteins that contain the domain.
2. There must be structures of at least three different peptide-bound complexes of the domain in the PDB (i.e., different domain Uniprots), and the peptides must all bind in the same location and orientation on the surface of the domain.
3. There must be at least three human proteins that contain the domain

4. We exclude repeat proteins (e.g., TPR repeats) that have evolved the ability to bind peptide multiple times in a manner consistent with convergent evolution.

We exclude domains for which peptide-binding includes catalytic modification of the peptide, which includes proteases and enzymes that add or remove post-translational modifications.

**Pfam–ligand interactions**. Domain–ligand interactions are calculated from the asymmetric units in the PDB. We define all non-polymer molecules except water in the PDB as ligands. A ligand interaction refers to at least one atomic contact with distance ≤4.5 Å between a protein domain and a ligand. Nucleic acids are treated in the same way as other ligands. Domain–DNA/RNA interfaces are calculated from the biological assemblies in the PDB. The interface with the largest number of Pfam HMM positions in contact with DNA or RNA is selected.

For each Pfam, we superpose all available domain structures based on their Pfam HMM positions and provide a script file to display the interactions in PyMOL. The domain structures and PyMOL script files are compressed as a single file, named after the Pfam ID (e.g., PK.tar.gz). The ligands are clustered based on the number of common Pfam HMM positions that they contact. One PyMOL session contains the coordinates of all structures for a given Pfam domain, and all interacting ligands. Each ligand and each group of ligands are defined as selection objects with the names of ligands; users can turn on or off each selection by clicking it in the object list. There are three types of coordinates provided by ProtCID: (i) the best domain with least missing coordinates of a PDB entry, marked by pdb (e.g., PK_pdb.tar.gz.); (ii) the best domain of a unique sequence, marked by unp (e.g., PK_unp.tar.gz); (iii) the best domain of a crystal form, marked by cryst (e.g., PK_cryst.tar.gz). The unp and cryst folders contain fewer domains, and these may be easier to open in PyMOL for big Pfam families. A user can download the gzipped tar files that contain the coordinate files of domain–ligand interactions along with PyMOL scripts from each Pfam page on the ProtCID web site (e.g., (http://dunbrack2.fccc.edu/ProtCiD/Results/EntityPfamArchWithPfam.aspx? PfamID=PK).

Besides the data for a Pfam and its interacting ligands, ProtCID also provide data sets for a ligand and its interacting Pfams. For each ligand, ProtCID provides domain coordinates and PyMOL scripts for structures of the Pfam with that ligand. All PyMOL scripts and structures are compiled and compressed into a single archive file named after this ligand (e.g., HEM.tar.gz) and can be downloaded from the page for each ligand (e.g., The coordinates of HEM-Pfams interactions can be downloaded from the web page (http://dunbrack2.fccc.edu/ProtCiD/IPdbfam/ PfamLigands.aspx?Ligand=HEM). In this way, users can obtain and view all homologous protein structures interacting with a specific ligand in one click.

**Protein–protein interactions**. A user can input a list of UniProt accession codes (in either form, e.g., BRAF_HUMAN or P15056) to identify interactions among them (http://dunbrack2.fccc.edu/ProtCiD/Search/Uniprots.aspx). There are two ways to identify interactions: "First to All" and "All to All". "First to All" refers to the interactions between the first protein in a list and all other proteins in the list (proteins 2 to $N$). These results may be of interest if the first protein is a hub protein with many protein interactors. "All to All" prompts the server to identify all structures that may occur among a list of input proteins, which is more useful for large protein complexes or complicated pathways with uncertain connections. A user can choose interface types to search for: either "Interfaces of Pfam domains" or "Interfaces of Input Sequences". In both cases, the server produces a Cytoscape network (http://www.cytoscape.org/) where each node is one of the input Uniprot identifiers. Edges are drawn between the nodes if there is structural information available for that pair of nodes.

The behavior is different depending on whether "Interfaces of Input Sequences" or "Interfaces of Pfam domains" is selected. When "Input sequences" is selected, clicking on each node provides a list of structures that contain that Uniprot sequence. An edge is labeled by the number of crystal forms (numerator) and the number of PDB entries (denominator) of the biggest interface cluster. Clicking on an edge provides a list of structures that contain direct interactions of the two Uniprot sequences connected by the edge. These can be useful in determining whether the target sequences are actually in common entries in the PDB.

By contrast, clicking on "Interfaces on Pfam domains" provides information on all structures that share Pfams with the input sequences. Clicking on the nodes provides a list of all structures that contain any of the Pfams in the node sequence. Clicking on an edge provides a list of all Pfam–Pfam clusters that connect the two nodes. If there are no Pfam–Pfam interactions identified from crystal structures and if one of the proteins has a professional peptide binding domain (PPBD), then an interaction is predicted and an edge is drawn. As an example, if we input the 5 Uniprot identifiers for the Cascade complex of the CRISPR-Cas system from *Thermobifida fusca*, ProtCID provides the interactions from structures of these specific proteins as well as and their Pfams (Supplementary Fig. 7).

Pfam assignments and interactions can be downloaded as text files. Like other queries, all coordinate files are downloadable for further study. Supplementary Fig. 8 shows the flowchart of the query of P53_HUMAN and its interactors. The list was collected from UniProt web page (https://www.uniprot.org/uniprot/ P04637#interaction).

**Pfam–Pfam interaction networks**. ProtCID also provides the ability to browse the interactions among Pfams without the need for input Uniprot sequence identifiers. A search for a single Pfam (e.g., Pkinase, http://dunbrack2.fccc.edu/ProtCiD/IPDBfam/PfamNetwork.htm?Pfam=Pkinase) produces a Cytoscape network with the input Pfam at the center, and edges to all Pfams that interact with the Pfam in PDB structures; the number of available structures is shown. It is also possible to browse networks of interacting Pfams (http://dunbrack2.fccc.edu/ProtCiD/Browse/PfamBioNet.aspx).

**Interactions of user-input sequences**. A user can input one or two amino acid sequences (instead of UniProt Identifiers) on the ProtCID web site to retrieve common interfaces between them or between homologs with the same Pfams (http://dunbrack2.fccc.edu/ProtCiD/Search/sequence.aspx and http://dunbrack2.fccc.edu/ProtCiD/Search/sequences.aspx). Sequences are searched against the Pfam HMM database by running HMMER 3.1 (http://hmmer.org/). A simple greedy algorithm is used to assign Pfams to the sequences. The hit with the best *E*-value (smallest value) and at least ≥80% coverage of the Pfam HMM is selected first. For each additional Pfam hit in order of *E*-value, if it does not overlap with any existing Pfam assignments by >10 residues on either end, then an assignment is made. After assigning Pfams, ProtCID returns interactions of Pfam architectures and individual Pfam domains.

## Data availability

The datasets generated during and/or analyzed during the current study are available at http://dunbrack2.fccc.edu/ProtCiD/paper/PaperDataDownload.htm. All ProtCID data are available via ProtCID web site (http://dunbrack2.fccc.edu/ProtCiD). ProtCID uses structures of the Protein Data Bank (https://www.rcsb.org/), sequences and UniProt IDs of UniProt (https://www.uniprot.org/), assemblies of Proteins, Interfaces, Structures and Assemblies (PISA, https://www.ebi.ac.uk/pdbe/pisa/), residue-level mapping between UniProt and PDB entries from Structure integration with function, taxonomy and sequence (SIFTS, https://www.ebi.ac.uk/pdbe/docs/sifts), and Pfam hidden Markov models (HMMs) from Pfam (ftp://ftp.ebi.ac.uk/pub/databases/Pfam).

## Code availablity

All source codes are written in C# and deposited in https://github.com/DunbrackLab/ProtCID. A demo program is provided in https://github.com/DunbrackLab/ProtCID_demo.

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

## Acknowledgements

This work was supported by NIH grant R35 GM122517.

## Author contributions

Q.X. conceived and designed the work, developed the project, analyzed the data, implemented the ProtCID web site, and wrote the manuscript. R.D. conceived and designed the work, analyzed the data, and wrote the manuscript.

## Competing interests

The authors declare no competing interests.
