## [Peer Review File · Nature Communications]

Reviewers' comments:

Reviewer #1 (Remarks to the Author):

The paper shows the new features of the ProtCID method, initially published in 2011 (with a previous publication in 2008) and especially focuses on applications of the method, showing a few possible ways to generate hypothesis with many examples for each of them. I find ProtCID to be a very interesting tool and an important contribution to the structural bioinformatics field and I have used it extensively in the past. The hypothesis presented are well researched, though I would not say that all of them are convincing.

I have detailed comments below per section, but I will first start with more general comments:

1) In my understanding this paper presents some new features of ProtCID compared to the already published papers. It would help the reader if that was briefly explained explicitly in a section in Results (to be placed at the beginning of Results), highlighting what are the main differences and new features (perhaps it can be added in the "ProtCID web site" section). There are some mentions in the introduction of what's new but I usually don't expect that a result is in the introduction.

2) Generally the writing and structuring of the paper should be streamlined. There are many places where it is difficult to follow and sometimes there are some sentences that seem to have been rushed out. Also the figures and legends need to be reviewed thoroughly and make as clear as possible. The paper presents a lot of content and is difficult to follow, so streamlining it would help readability. For instance I find the abstract and introduction too long and detailed, e.g. I would not present the examples in the introduction.

3) A general criticism of something that the authors do not justify is why the homology and clustering is done at the PFAM level. In terms of methodology that is sound, but in terms of comparing and clustering interfaces I have my doubts that finding similarities/dissimilarities in interfaces of distantly related proteins or domains is going to provide the best signal in all situations. It is well known that interfaces and more generally quaternary structure diverge between homologous proteins of the same family or even for the same protein under different conditions. Putting it in other words: protein tertiary structure is far better conserved than quaternary structure. See for instance the work of Eileen Jaffe (e.g. <https://www.ncbi.nlm.nih.gov/pubmed/22182754>). Could the authors comment on that?

Abstract

"it is virtually impossible for a scientist who is not trained in structural bioinformatics to access this information across all of the structures that are available of any one extensively studied protein or protein family" -> Should be softened. There are many tools available for this for biologists

"We present ProtCID (the Protein Common Interface Database) as a webserver and database that makes comprehensive, PDB-wide structural information on the interactions of proteins and individual protein domains with other molecules accessible to scientists at all levels. " -> Difficult to read, fix punctuation or the way that is expressed

"One example is a homodimer of HRAS, KRAS, and NRAS..." -> Not clear what this means. Why is this in abstract, either make a point or leave it for the results section

Introduction

Overall comment: in my opinion the Introduction is too verbose and sometimes it repeats similar

ideas. I'd suggest to the authors to streamline it so that it becomes easier to understand to the reader and so that the points that are trying to be introduced are clearer. There's no need to explain half of the results in the introduction, or at least they can be explained more briefly. Perhaps simply leaving all the specific examples out and instead just presenting the different mechanisms for hypothesis generation would improve it. The specific examples of the different biological systems thoroughly studied can come in Results.

Page 3.

Please cite the PDB, e.g. Burley 2018,
<https://academic.oup.com/nar/article/47/D1/D464/5144139>

"The number of structures for a protein and its homologues can reach into the hundreds or thousands" -> The full PDB is 150,000 entries right now. I don't see how can a single family have hundreds of thousands of structures.

"It is virtually impossible when there are dozens or hundreds of available structures." -> As in abstract, please soften. There are many good tools that cover many aspects of this.

Page 4.

"This is in contrast to the asymmetric unit, which is the set of coordinates used to model the unit cell and the crystal lattice when copied and placed with rotational and translational symmetry operators." -> Not well expressed

"(i.e., made from parts or all of multiple copies of the ASU)" -> I'd rephrase to "made from multiple copies of the ASU or parts of it"

"Various authors have estimated the accuracy of the biological assemblies in the PDB in the range of 80-90%" -> Please cite also Baskaran et al 2014 (<https://bmcstructbiol.biomedcentral.com/articles/10.1186/s12900-014-0022-0>) and Levy 2007 (<https://www.ncbi.nlm.nih.gov/pubmed/17997962>)

"especially when the proteins in the different crystals are homologous but not identical" -> Why "especially"? This applies equally well to both identical or homologous proteins

Page 5

"ProtCID allows users to download coordinates the PyMol scripts for visualizing all available interfaces" -> Does not make sense, needs to be completed or rephrased

Page 7

"...and input to ProtCID" -> Rephrasing suggestion: "and used as inputs to ProtCID"

Results

For clarity and readability I'd suggest separating in subsections the "Generating hypotheses for oligomeric protein assemblies with ProtCID" section. One subsection per topic: EGFR domain, RAS, Bromodomains.

A suggestion that should improve readability: would it be possible to provide links to the relevant ProtCID page for a cluster whenever a cluster is mentioned in the text? That will make it easier to follow the points (for instance the reader could go and download a PyMol file) and would provide

more visibility for the server, while showing a possible way of using it.

Page 9

"asymmetric or heterologous dimers have the risk of forming polymer chains as true polymeric chains do such as actin" -> I would use the word "fiber" in favour of "true polymeric chain"

"A total of 100 of 104 " -> "A total of 100 out of 104"

"EPPIC predicts only the heterodimers as biological assemblies; it predicts that all of the homodimers in this ProtCID cluster are monomers" ->Just a clarification: if one looks purely at the interface level, EPPIC does detect a clear biological signal for the relevant heterologous interface in some of the cases (e.g. PDB id 2gs2, interface 1, <http://eppic-web.org/ewui/#interfaces/2gs2>), however EPPIC is strict about never predicting asymmetric assemblies and thus at the assembly prediction level the proteins are considered monomers. So in terms of evolutionary conservation of the residues at the interface the signal is clear. The Duarte et al 2012 paper (<https://bmcbioinformatics.biomedcentral.com/articles/10.1186/1471-2105-13-334>) actually uses this as a highlight example to show how sequence conservation can detect this biologically relevant interface.

Page 11

The story as it is exposed is not easy to follow. It would help the reader a lot if the authors explicitly say what is the active state (I suppose the GTP bound) and the inactive state (I suppose GDP bound). My reading is that 88% of the structures containing the dimer are in active form, whilst 50% of structures without the dimer are in the active form. That can indeed originate the hypothesis but it is not strong evidence at all. In any case the point should be explained more clearly, perhaps with the help of figures or tables.

Page 12

To illustrate the Ras/Ras effector binding story, the authors refer in first paragraph of page 12 to Figure S2 d) and e). I could only find panels a) b) and c) for the provided Supplementary figure 2. I am guessing they refer to panels b) and c). Please correct this, it is quite hard to follow and do the guesswork. Additionally the legend of Figure S2c does not make a lot of sense to me. What's shown is not heterotetramers as the legend says, or am I missing something? In general I am getting lost to understand what is the point that the authors try to make. Specifically what do they mean by "the partners could bind the alpha4-alpha5 dimer" and additionally why is that significant? Please I'd advise the authors to elaborate it better.

In the bromodomain story, the authors call the dimer a "head-to-tail" symmetric dimer. They should point to Figure S3 when mentioning this the first time, so as to make clear what "head-to-tail" means. When reading this I got confused by nomenclature used by Levy (see Levy 2008, <https://www.nature.com/articles/nature06942>) where isologous=face-to-face and heterologous=face-to-back.

Page 23.

Modeling of protein complexes in ProtCID: in general I like the idea of modelling protein complexes out of interfaces found in many crystal forms. However this idea presents a few challenges that I don't see addressed in the manuscript:

- 1) How can it be explained that the modelled homo-oligomers are not in any of the crystals? I.e. why is it that not a single crystal contain all relevant interfaces together?
- 2) The modelling requires some quality numbers attached to it. One thing is approximately modelling a tetramer out of 2 interfaces, another is seeing how reasonable the model is: what's

the interface area? Are there many clashes? Are there any domain movements or flexibility assumed when creating the models? Was the modelling purely manual or were some software docking methods used?

In Figure 6, the legend refers within panel a) to subpanels 1, 2, 3. However the subpanel labels are not in the figure which needs some guesswork to understand.

Page 29

"ProtCID can be used to obtain define larger oligomeric assemblies" -> Does not make sense, please review

Signed: Jose M. Duarte

Reviewer #2 (Remarks to the Author):

The paper seeks to simplify the study of domain interactions within and between proteins, as well as among proteins and peptides, nucleic acids, and ligands, regardless of one's training in structural informatics. As a result, the new tool and website described in the paper are extremely significant. They enable any researcher to discover new potential interactions that then guide appropriate experiments. A further significant benefit is the capability to find biological assemblies already present in the PDB but not annotated as such by the original authors, or by competing external tools, such as PISA. To achieve this, ProtCID introduces a series of innovative annotation techniques for the analysis of Pfam domains in the PDB structure databank and their Uniprot identifiers. These annotations are combined with calculations of distance between atoms in the structures and/or the surface area of potential interfaces to cluster proteins into groups which are believed to be interacting in similar ways. Several examples show the power of this new approach to template matching. Domain-domain clustering suggested a homodimer of ErbB that was confirmed by experimentation. So were a homodimer of Ras, two homodimers of BET Bromodomains, and a peptide substrate of a BET Bromodomain dimer. This same process also allowed the authors to take known homodimers of previously determined activity and then extend their annotation to numerous other proteins including the human BRAF, CSK, ITK, MLKL, RPK2, and RAF1 proteins as well as a number of non-human proteins, and domain-domain or Pfam-Pfam interfaces found across a wide number of PDB structures. Importantly, this story was repeated for peptide, protein, nucleic acid, and ligand binding and it reveals many PDB entries likely to bind specific types of targets, many of which have been previously supported in the literature and others which seem very promising to investigate. The protein structural clustering performance of ProtCID appears markedly superior in comparison to the PISA and to machine learning techniques. A noteworthy example is the difficult problem of identifying asymmetric assemblies, which is beyond common biophysical and covariation methods.

The most impressive aspect of this manuscript and the provided tool are its success at several levels of scale and application. Not only can this tool be applied on a chain by chain basis, but it can also be applied on the more specific Pfam domain level. The fact that applying clustering at these two scales allows for interrogation of so many potential interactions of interest makes the case presented in this manuscript as well as the tool itself very appealing. A wide range of researchers will be able to make use of this tool, whether they are attempting to uncover the potential structure of an oligomer or assembly that their protein of interest participates in, or its potential binding interactions with other proteins, peptides, nucleic acids, or ligands, this tool can be of help. This and the strong introduction explaining the need for and potential usefulness of such a tool make for a good paper.

A possible area of improvement might be the anecdotal nature of many of the results and also some more careful editing for clarity. Would there be a way to deploy and test this tool for a larger-scale validation. Perhaps this could be attempted at least for one type of ligand? On the other hand it may make sense to present the work as is, make the tool available to the community and leave for follow up papers large scale validation. Another question that arises is whether there is any possible scoring specific to clusters or their constituents that could help experimentalists prioritize their follow-up work. This would also make it possible to examine where in a ranked list of cluster constituents those PDB entries which are annotated by the original authors or by PISA fall, making it possible to perform analyses like Area Under the Receiver Operating Curve (AUROC) or other ranking metrics.

In summary, this paper represents a qualitative advance in the annotation of function based on structure and goes some way to fulfilling the promise of Structural Genomics, which was to solve structure in order to understand functionality. Aside from the Minor edits suggest above and below, the work is important, well done, clearly described and leads to tools that will be widely used by the community. I recommend acceptance.

Minor Edits

Edits for clarity are mentioned in list form below:

- Page 5: "ProtCID allows user to download coordinate the PyMol scripts for visualizing all available interfaces."
- Page 9: Consider rewording or making two sentences "This dimer was unexpected because most protein homodimers are symmetric or "isologous", and asymmetric or "heterologous" dimers have the risk of forming polymer chains as true polymeric chains do such as actin' EGFR probably cannot do this because of crowding at the membrane."
- Page 12: A reference is made to "Supplementary Figure 2e" however Supplementary Figure 2 only has panels a, b, and c.
- Page 13: A reference is made to "Supplementary Figure 3b", while Supplementary Figure 3 does have captions for panel a and b, the figure does not have labels for either.
- Page 19: "Figure 3a" is referenced in bold, but all other figure references are not bolded, unless this has particular significance it should be kept consistent.
- Page 22: In the caption for Figure5, "with" in the final sentence should be "which"
- Page 29: Please choose "ProtCID can be used to obtain/define larger oligomeric assemblies, which we demonstrated for ..."
- Page 30: Please correct "All downloadable structure files in ProtCID come with different PyMOL scripts which can be just double clicked to open in PyMOL." Also thank you for providing this feature it is very useful!
- Page 41: The first link in the section "Interactions of user sequences" is not functional (<http://dunbrack2.fccc.edu/ProtCiD/Search/sequence.aspx>) please remove or update it.
- Supplementary Table 1: Please clarify "..., if the domains can come any chain architectures that contain them."
- Supplementary Table 7: Why is "Asp" highlighted?

Our response to the reviewers' comments are shown in blue type and prefixed with two asterisks. We have substantially rewritten the paper in response to the reviews. The original paper was submitted to *Nature Methods* and we understand that a *Nature Communications* paper has somewhat different aims. We focused on the major change in ProtCID, which is to provide clusters of interacting molecules at the protein domain-level and how this change enables new opportunities for developing hypotheses and providing analysis on protein-protein interactions and on protein/ligand, protein/peptide, and protein/nucleic-acid interactions. We have rewritten the Discussion which previously reviewed the results section in too much detail. Now we highlight the important features of ProtCID that are often not present in other servers of this type. We did not specifically mention each server that has each problem since they would end up adding another dozen references and we are not trying to single out specific servers. Some of the commonly used servers (3DID, IBIS) have several of the problems described. We have marked changes in the manuscript in red if they are substantial insertions of new text or if there are changes that specifically respond to a comment of one of the reviewers.

Reviewers' comments:

Reviewer #1 (Remarks to the Author):

The paper shows the new features of the ProtCID method, initially published in 2011 (with a previous publication in 2008) and especially focuses on applications of the method, showing a few possible ways to generate hypothesis with many examples for each of them. I find ProtCID to be a very interesting tool and an important contribution to the structural bioinformatics field and I have used it extensively in the past. The hypothesis presented are well researched, though I would not say that all of them are convincing.

I have detailed comments below per section, but I will first start with more general comments:

1) In my understanding this paper presents some new features of ProtCID compared to the already published papers. It would help the reader if that was briefly explained explicitly in a section in Results (to be placed at the beginning of Results), highlighting what are the main differences and new features (perhaps it can be added in the "ProtCID web site" section). There are some mentions in the introduction of what's new but I usually don't expect that a result is in the introduction.

** We have rearranged the manuscript to emphasize the major change in ProtCID which is to analyze interactions at the domain level rather than the full chain level. We have tried to make this clear in the Introduction and abstract and at the beginning of the Results section.

2) Generally the writing and structuring of the paper should be streamlined. There are many places where it is difficult to follow and sometimes there are some sentences that seem to have been rushed out. Also the figures and legends need to be reviewed thoroughly and make as clear as possible. The paper presents a lot of content and is difficult to follow, so streamlining it would help readability. For instance I find the abstract and introduction too long and detailed, e.g. I would not present the examples in the introduction.

** We have rearranged the paper as described above. Part of the problem is that we pitched it to *Nature Methods* as a method for developing hypotheses on protein interactions, and that approach made some examples in the initial submission a bit strained. The paper is better organized. We have removed the explicit examples from the introduction and the discussion. We have reviewed and edited the captions to make them clearer.

3) A general criticism of something that the authors do not justify is why the homology and clustering is done at the PFAM level. In terms of methodology that is sound, but in terms of comparing and clustering interfaces I have my doubts that finding similarities/dissimilarities in interfaces of distantly related proteins or domains is going to provide the best signal in all situations. It is well known that interfaces and more generally quaternary structure diverge between homologous proteins of the same family or even for the same protein under different conditions. Putting it in other words: protein tertiary structure is far better conserved than quaternary structure.

See for instance the work of Eileen Jaffe (e.g. <https://www.ncbi.nlm.nih.gov/pubmed/22182754>). Could the authors comment on that?

** We agree that for some protein families, the interactions of domains can be quite different at long evolutionary distances. It is why we do not perform clustering at the Pfam clan (superfamily) level. However, the advantage of the ProtCID approach is that *if* we do find a cluster at the Pfam level (whether by chains or by domains), and the cluster contains multiple crystal forms and homologous but non-identical proteins, then it is likely to be biological. If the domain/domain interactions have changed in a particular family, then they will not cluster together or they may appear in different clusters with no Uniprot sequences in common. For instance, kinase domains dimerize in several different ways, but these are found in different clusters with no Uniprot in common (e.g. the BRAF/MLKL/RIPK2/CSK/ITK cluster vs the EGFR/ErbB2/ErbB3/ErbB4 cluster). On the other hand, there are large ProtCID protein-protein cluster that contain proteins with pairwise sequence identities less than 10%. We use HMM/HMM comparisons to assign Pfams, so we sometimes identify remote member of some families. We have visualized 100s of different clusters, and have not found structures that do not belong to a family or interfaces that do not resemble the other interfaces in the cluster. It is more likely that on occasion an interface is missed because either the sequence does not align to the HMM very well or the interface is sufficiently distorted that the Q score cannot detect the similarity.

** It is also true that the same protein (or proteins) can have different domain/domain interactions under different conditions (as my Fox Chase colleague Eileen Jaffe has pointed out; her work was in part a motivation for us to examine biological assemblies in the PDB and to develop ProtCID). These would appear in different ProtCID clusters, which is in fact very useful in identifying which structures are in which state. We provide an example of two intrachain Pfam/Pfam domain interactions for Adenylate enzymes. The two conformations are two different steps in the catalytic cycle. ProtCID enables the rapid identification of the structures in the two states.

Abstract

"it is virtually impossible for a scientist who is not trained in structural bioinformatics to access this information across all of the structures that are available of any one extensively studied protein or protein family" -> Should be softened. There are many tools available for this for biologists

** We softened this comment. But it is true that there are few (if any) tools that a) cluster interactions; b) provide downloads of coordinates of all members in one click (plenty provide one structure per click, which is tedious); c) build out crystals instead of relying on the ASU or assemblies deposited in the PDB. I think that downloading all the coordinates in one click (with annotations) is rare, but it is a crucial task in analyzing structural data. In addition, the other tools have one or more problems: 1) they may be years out of date; 2) they don't provide protein names or Uniprot accession codes (they just give PDB codes); 3) they don't provide ways to search by homology (which our Pfam assignments enable); 4) they are focused on just one aspect of protein interactions (protein-protein interactions or protein/peptide or protein/ligand interactions); 5) I don't think there is *any* tool that counts the number of crystal forms available for a protein domain family and the number of crystal forms for a cluster of protein-protein interactions; this information is critical for providing some evidence in favor (or against) the likely biological relevance of an interaction observed in crystals.

"We present ProtCID (the Protein Common Interface Database) as a webserver and database that makes comprehensive, PDB-wide structural information on the interactions of proteins and individual protein domains with other molecules accessible to scientists at all levels. " -> Difficult to read, fix punctuation or the way that is expressed

** Fixed.

"One example is a homodimer of HRAS, KRAS, and NRAS..." -> Not clear what this means. Why is this in abstract, either make a point or leave it for the results section

** Removed from abstract. We wanted to highlight the RAS homodimer since RAS is of such interest in cancer (and we work in a cancer center) but it does not belong in the abstract.

Introduction

Overall comment: in my opinion the Introduction is too verbose and sometimes it repeats similar ideas. I'd suggest to the authors to streamline it so that it becomes easier to understand to the reader and so that the points that are trying to be introduced are clearer. There's no need to explain half of the results in the introduction, or at least they can be explained more briefly. Perhaps simply leaving all the specific examples out and instead just presenting the different mechanisms for hypothesis generation would improve it. The specific examples of the different biological systems thoroughly studied can come in Results.

** We streamlined the intro (now <1000 words). Sometimes it is worth highlighting the major results in a paper in the intro because people don't read the whole paper. It was more relevant to the initial submission where we were emphasizing "hypothesis generation" with ProtCID.

Page 3.

Please cite the PDB, e.g. Burley 2018, <https://academic.oup.com/nar/article/47/D1/D464/5144139>

** Of course. I should know better 😊.

"The number of structures for a protein and its homologues can reach into the hundreds or thousands" -> The full PDB is 150,000 entries right now. I don't see how can a single family have hundreds of thousands of structures.

** It was easy to misread: The sentence read "100s or 1000s", not "100s of 1000s"; we changed it to "100s or even 1000s" to make it clearer.

"It is virtually impossible when there are dozens or hundreds of available structures." -> As in abstract, please soften. There are many good tools that cover many aspects of this.

** We softened this. But there are not many tools that cluster interaction information *and* provide for downloading all the structures in a particular protein family and superposing them.

Page 4.

"This is in contrast to the asymmetric unit, which is the set of coordinates used to model the unit cell and the crystal lattice when copied and placed with rotational and translational symmetry operators." -> Not well expressed

** We shortened it to this: "This is in contrast to the asymmetric unit, which is the set of coordinates used to model the crystal lattice."

"(i.e., made from parts or all of multiple copies of the ASU)" -> I'd rephrase to "made from multiple copies of the ASU or parts of it"

** This sentence was removed.

"Various authors have estimated the accuracy of the biological assemblies in the PDB in the range of 80-90%" -> Please cite also Baskaran et al 2014 (<https://bmcsstructbiol.biomedcentral.com/articles/10.1186/s12900-014-0022-0>) and Levy 2007

<https://www.ncbi.nlm.nih.gov/pubmed/17997962>)

** Done.

"especially when the proteins in the different crystals are homologous but not identical" -> Why "especially"?
This applies equally well to both identical or homologous proteins

** We showed in our 2008 benchmarking paper that clusters were more predictive of true biological interactions if they contained non-identical proteins (at least two with less than 90% pairwise sequence identity) than if the proteins were 100% identical. It does occur that if a cluster contains identical proteins, even in a number of crystal forms, it may be because the protein likes to form the dimer at high concentrations but the dimer is not biologically relevant for any kind of oligomeric assembly. The famous example is of the T4 lysozyme homodimer that is found in 21 different crystal forms and 501 entries. When the proteins in a cluster are non-identical, very similar crystallization-induced contacts are less likely to occur in different crystal forms. Of course when there are 100s of different crystal forms (e.g. for Pkinase), there may be contacts that occur for different proteins that are quite similar by coincidence. This is why we report the total number of crystal forms that are available for any Pfam architecture as well as the number of crystal forms that occur for each cluster.

Page 5

"ProtCID allows users to download coordinates the PyMol scripts for visualizing all available interfaces" ->
Does not make sense, needs to be completed or rephrased

** The sentence was missing the word "and." Fixed.

Page 7

"...and input to ProtCID" -> Rephrasing suggestion: "and used as inputs to ProtCID"

** Done; although "input" is OK as an English verb:
https://www.macmillandictionary.com/us/dictionary/american/input_2

Results

For clarity and readability I'd suggest separating in subsections the "Generating hypotheses for oligomeric protein assemblies with ProtCID" section. One subsection per topic: EGFR domain, RAS, Bromodomains.

** Each topic is only a paragraph or two which would make the subsections pretty short. We will let the editors at *Nature Communications* help us with this. Right now, we shortened them, and put all the weak dimers together in one section after the section on the advantages of domain-level analysis for protein-protein interactions.

A suggestion that should improve readability: would it be possible to provide links to the relevant ProtCID page for a cluster whenever a cluster is mentioned in the text? That will make it easier to follow the points (for instance the reader could go and download a PyMol file) and would provide more visibility for the server, while showing a possible way of using it.

** We added this where possible. It depends on whether the journal will allow us to put in the links. This will be fixed during the publication stage, if the journal accepts the article. We put all PyMol sessions and Excel data files for figures, supplementary figures and supplementary tables on <http://dunbrack2.fccc.edu/ProtCID/paper/PaperDataDownload.htm> for downloading. We indicated it in the "Data Availability" section.

Page 9

“asymmetric or heterologous dimers have the risk of forming polymer chains as true polymeric chains do such as actin” -> I would use the word “fiber” in favour of “true polymeric chain”

** We changed it to “filamentous.”

“A total of 100 of 104 “ -> “A total of 100 out of 104”

** Done.

“EPPIC predicts only the heterodimers as biological assemblies; it predicts that all of the homodimers in this ProtCID cluster are monomers” ->Just a clarification: if one looks purely at the interface level, EPPIC does detect a clear biological signal for the relevant heterologous interface in some of the cases (e.g. PDB id 2gs2, interface 1, <http://eppic-web.org/ewui/#interfaces/2gs2>), however EPPIC is strict about never predicting asymmetric assemblies and thus at the assembly prediction level the proteins are considered monomers. So in terms of evolutionary conservation of the residues at the interface the signal is clear. The Duarte et al 2012 paper (<https://bmcbioinformatics.biomedcentral.com/articles/10.1186/1471-2105-13-334>) actually uses this as a highlight example to show how sequence conservation can detect this biologically relevant interface.

** We clarified this example – the reviewer is correct that the interface is predicted to be biological but that the dimer is not predicted to be a biological assembly in order to avoid identifying filamentous assemblies in crystals as biological, most of which are not biological.

Page 11

The story as it is exposed is not easy to follow. It would help the reader a lot if the authors explicitly say what is the active state (I suppose the GTP bound) and the inactive state (I suppose GDP bound). My reading is that 88% of the structures containing the dimer are in active form, whilst 50% of structures without the dimer are in the active form. That can indeed originate the hypothesis but it is not strong evidence at all. In any case the point should be explained more clearly, perhaps with the help of figures or tables.

** This is tricky because there is not a one-to-one correspondence between GTP/GDP and active/inactive structures of RAS proteins. We deliberately did not try to classify the structures as active or inactive since this would be a major project (which we are undertaking separately). Instead, we just calculated the numbers with GTP (or triphosphate analogs) vs GDP. We added this sentence: “Most structures with GTP are considered “active”, while most structures with GDP are “inactive.” “

Page 12

To illustrate the Ras/Ras effector binding story, the authors refer in first paragraph of page 12 to Figure S2 d) and e). I could only find panels a) b) and c) for the provided Supplementary figure 2. I am guessing they refer to panels b) and c). Please correct this, it is quite hard to follow and do the guesswork. Additionally the legend of Figure S2c does not make a lot of sense to me. What’s shown is not heterotetramers as the legend says, or am I missing something? In general I am getting lost to understand what is the point that the authors try to make. Specifically what do they mean by “the partners could bind the alpha4-alpha5 dimer” and additionally why is that significant? Please I’d advise the authors to elaborate it better.

** We fixed all of this. We added the RAS/RA Pfam-Pfam dimer cluster to the main paper. After showing the RAS homodimers, we show the tetramers that can form with RAS effector domains. In the original submission, we only showed the trimers (two copies of HRAS and one effector protein), when we should have shown the HRAS homodimer with one effector bound to each RAS, i.e. a heterotetramer. The significance is that if the alpha4-alpha5 dimer is a biologically active form, it is consistent with binding RAS-effector domains. This is in contrast to the so-called beta dimer, which cannot bind these proteins.

In the bromodomain story, the authors call the dimer a “head-to-tail” symmetric dimer. They should point to Figure S3 when mentioning this the first time, so as to make clear what “head-to-tail” means. When reading this I got confused by nomenclature used by Levy (see Levy 2008, <https://www.nature.com/articles/nature06942>) where isologous=face-to-face and heterologous=face-to-back.

** We added a reference to Figure 3e when we first mention head-to-tail. “Head” and “Tail” refer to the long axis of the protein (as shown in Figure 3). They are both C2-symmetric “face to face” dimers. It’s just that in the head-to-tail dimers the rotation axis is perpendicular to the long axis. In the head-to-head dimers, the axis is parallel to the long axis (as it would be if two people stood head to head).

Page 23.

Modeling of protein complexes in ProtCID: in general I like the idea of modelling protein complexes out of interfaces found in many crystal forms. However this idea presents a few challenges that I don’t see addressed in the manuscript:

1) How can it be explained that the modelled homo-oligomers are not in any of the crystals? I.e. why is it that not a single crystal contain all relevant interfaces together?

** This can occur in two different circumstances: 1) we can model the homooligomer of a multi-domain protein when there is no structure of the full-length protein, but there are pairwise domain-domain structures (e.g. for a three-domain structure, there are structures of domains 1-2 and domains 2-3 but none of 1-2-3); 2) there is no structure of the specific protein of interest (e.g. the human protein, when the full-length structures are from some other species) but some human domain structures are known; 3) when there is no structure of a complex of several different proteins, but there are individual structures of the pairwise heterodimeric structures. We are not trying to present a general modeling strategy (or software) for large homo- or heterooligomeric complexes in this paper. We have removed most of the commentary on the modeling aspects, which are not really the point of the paper.

2) The modelling requires some quality numbers attached to it. One thing is approximately modelling a tetramer out of 2 interfaces, another is seeing how reasonable the model is: what’s the interface area? Are there many clashes? Are there any domain movements or flexibility assumed when creating the models? Was the modelling purely manual or were some software docking methods used?

** We now present only one model – the human HBO1 complex based on the yeast HBO1 complex. For two of the proteins in the complex, there are experimental structures of the individual human proteins, and these proteins superpose pretty well on the yeast proteins in the heterotetrameric structure. For the other two, we used Swissmodel to make models of the human proteins from the yeast proteins in the same yeast HBO1 structures. The models also superpose pretty well with few clashes. The model is only an illustrative example, and more detailed analysis of the sequence alignment and the clashes would be more than is required here.

In Figure 6, the legend refers within panel a) to subpanels 1, 2, 3. However the subpanel labels are not in the figure which needs some guesswork to understand.

** This section has been rearranged.

Page 29

“ProtCID can be used to obtain define larger oligomeric assemblies” -> Does not make sense, please review

** Sentence removed in the revision.

Signed: Jose M. Duarte

Reviewer #2 (Remarks to the Author):

The paper seeks to simplify the study of domain interactions within and between proteins, as well as among proteins and peptides, nucleic acids, and ligands, regardless of one's training in structural informatics. As a result, the new tool and website described in the paper are extremely significant. They enable any researcher to discover new potential interactions that then guide appropriate experiments. A further significant benefit is the capability to find biological assemblies already present in the PDB but not annotated as such by the original authors, or by competing external tools, such as PISA. To achieve this, ProtCID introduces a series of innovative annotation techniques for the analysis of Pfam domains in the PDB structure databank and their Uniprot identifiers. These annotations are combined with calculations of distance between atoms in the structures and/or the surface area of potential interfaces to cluster proteins into groups which are believed to be interacting in similar ways. Several examples show the power of this new approach to template matching. Domain-domain clustering suggested a homodimer of ErbB that was confirmed by experimentation. So were a homodimer of Ras, two homodimers of BET Bromodomains, and a peptide substrate of a BET Bromodomain dimer. This same process also allowed the authors to take known homodimers of previously determined activity and then extend their annotation to numerous other proteins including the human BRAF, CSK, ITK, MLKL, RPK2, and RAF1 proteins as well as a number of non-human proteins, and domain-domain or Pfam-Pfam interfaces found across a wide number of PDB structures. Importantly, this story was repeated for peptide, protein, nucleic acid, and ligand binding and it reveals many PDB entries likely to bind specific types of targets, many of which have been previously supported in the literature and others which seem very promising to investigate. The protein structural clustering performance of ProtCID appears markedly superior in comparison to the PISA and to machine learning techniques. A noteworthy example is the difficult problem of identifying asymmetric assemblies, which is beyond common biophysical and covariation methods.

The most impressive aspect of this manuscript and the provided tool are its success at several levels of scale and application. Not only can this tool be applied on a chain by chain basis, but it can also be applied on the more specific Pfam domain level. The fact that applying clustering at these two scales allows for interrogation of so many potential interactions of interest makes the case presented in this manuscript as well as the tool itself very appealing. A wide range of researchers will be able to make use of this tool, whether they are attempting to uncover the potential structure of an oligomer or assembly that their protein of interest participates in, or its potential binding interactions with other proteins, peptides, nucleic acids, or ligands, this tool can be of help. This and the strong introduction explaining the need for and potential usefulness of such a tool make for a good paper.

A possible area of improvement might be the anecdotal nature of many of the results and also some more careful editing for clarity. Would there be a way to deploy and test this tool for a larger-scale validation. Perhaps this could be attempted at least for one type of ligand? On the other hand it may make sense to present the work as is, make the tool available to the community and leave for follow up papers large scale validation.

** We have previously validated the protein-protein complexes on the chain level (Xu et al 2008); it is not likely that benchmarking on the domain level would be more informative, and in fact would be very hard to do because the validated biological assemblies are mostly single-domain proteins. It is possible to perform validation at the clan level, e.g. how do different members of the same Pfam bind peptides or nucleic acids, but we view this as beyond the scope of the current paper and a task for future work. The clans in Pfam are problematic in some cases, since they represent different amounts of structure (e.g. some Pfams in a clan represent repeated copies of a domain while other members consist of just one copy; some Pfams contain insertions of domains that are not present in other Pfams in the same clan). We view ProtCID as a window on the available structural information for each Pfam domain, and a tool for generating hypotheses of how any given protein that contains the Pfam might interact with other molecules.

Another question that arises is whether there is any possible scoring specific to clusters or their constituents that could help experimentalists prioritize their follow-up work. This would also make it possible to examine

where in a ranked list of cluster constituents those PDB entries which are annotated by the original authors or by PISA fall, making it possible to perform analyses like Area Under the Receiver Operating Curve (AUROC) or other ranking metrics.

** This would be very difficult to accomplish, since we would be analyzing the contributions of 100s of authors to structures in the PDB. We are deliberately somewhat agnostic about whether a cluster observed in ProtCID demonstrates biologically relevant interactions. Rather, we present the data and allow users to utilize the information to develop a hypothesis which they are then free to test experimentally. There is not enough experimental data on verified biological assemblies to assign probabilities as a function of: 1) sequence diversity; 2) surface area; 3) number of crystal forms in the cluster; 4) number of crystal forms in total; 5) number of Pfam architectures, etc.

In summary, this paper represents a qualitative advance in the annotation of function based on structure and goes some way to fulfilling the promise of Structural Genomics, which was to solve structure in order to understand functionality. Aside from the Minor edits suggest above and below, the work is important, well done, clearly described and leads to tools that will be widely used by the community. I recommend acceptance.

Minor Edits

Edits for clarity are mentioned in list form below:

- Page 5: "ProtCID allows user to download coordinate the PyMol scripts for visualizing all available interfaces."

** Fixed.

- Page 9: Consider rewording or making two sentences "This dimer was unexpected because most protein homodimers are symmetric or "isologous", and asymmetric or "heterologous" dimers have the risk of forming polymer chains as true polymeric chains do such as actin' EGFR probably cannot do this because of crowding at the membrane."

** Fixed

- Page 12: A reference is made to "Supplementary Figure 2e" however Supplementary Figure 2 only has panels a, b, and c.

** Fixed.

- Page 13: A reference is made to "Supplementary Figure 3b", while Supplementary Figure 3 does have captions for panel a and b, the figure does not have labels for either.

** Fixed

- Page 19: "Figure 3a" is referenced in bold, but all other figure references are not bolded, unless this has particular significance it should be kept consistent.

** Fixed

- Page 22: In the caption for Figure5, "with" in the final sentence should be "which"

** This example has been removed.

- Page 29: Please choose “ProtCID can be used to obtain/define larger oligomeric assemblies, which we demonstrated for ...”

** Removed.

- Page 30: Please correct “All downloadable structure files in ProtCID come with different PyMOL scripts which can be just double clicked to open in PyMOL.” Also thank you for providing this feature it is very useful!

** We removed the redundant reference to PyMOL. We are glad it is useful.

- Page 41: The first link in the section “Interactions of user sequences” is not functional (<http://dunbrack2.fccc.edu/ProtCiD/Search/sequence.aspx>) please remove or update it.

** This works now.

- Supplementary Table 1: Please clarify “..., if the domains can come any chain architectures that contain them.”

** Removed.

- Supplementary Table 7: Why is “Asp” highlighted?

** Fixed

REVIEWERS' COMMENTS:

Reviewer #1 (Remarks to the Author):

The authors have made substantial efforts to rewrite much of the paper. It is very much appreciated. All my comments have been addressed satisfactorily. I think the paper has gained in clarity and readability, the main points are now more prominently highlighted and it is easier to follow. I especially like the new Introduction and Discussion sections. The paper represents an important addition to the structural bioinformatics field, extracting hidden information out of protein crystals, which elsewhere is mostly disregarded. I think the paper will be welcomed not only by the structural biology community but by the wider biomedical community.